# Fish primary embryonic pluripotent cells assemble into retinal tissue mirroring in vivo early eye development

**Lucie Zilova[1†], Venera Weinhardt[1†], Tinatini Tavhelidse[1], Christina Schlagheck[1,2], Thomas Thumberger[1], Joachim Wittbrodt[1]\***

[1]Centre for Organismal Studies Heidelberg, Heidelberg University, Heidelberg, Germany; [2]Heidelberg International Biosciences Graduate School HBIGS and HeiKa Graduate School on "Functional Materials", Heidelberg, Germany

**Abstract** Organoids derived from pluripotent stem cells promise the solution to current challenges in basic and biomedical research. Mammalian organoids are however limited by long developmental time, variable success, and lack of direct comparison to an in vivo reference. To overcome these limitations and address species-specific cellular organization, we derived organoids from rapidly developing teleosts. We demonstrate how primary embryonic pluripotent cells from medaka and zebrafish efficiently assemble into anterior neural structures, particularly retina. Within 4 days, blastula-stage cell aggregates reproducibly execute key steps of eye development: retinal specification, morphogenesis, and differentiation. The number of aggregated cells and genetic factors crucially impacted upon the concomitant morphological changes that were intriguingly reflecting the in vivo situation. High efficiency and rapid development of fish-derived organoids in combination with advanced genome editing techniques immediately allow addressing aspects of development and disease, and systematic probing of impact of the physical environment on morphogenesis and differentiation.

**\*For correspondence:**
jochen.wittbrodt@cos.uni-heidelberg.de

[†]These authors contributed equally to this work

**Competing interests:** The authors declare that no competing interests exist.

## Introduction

Organ development is a complex process of orchestrated events of tissue specification, morphogenesis, and differentiation in specialized cell types. In vertebrates, retinal development is initiated during early neurulation, when the eye field is specified by the coordinated expression of eye field transcription factors Rx, Lhx2, Pax6, Six3, and Otx2 within the region of the anterior neural plate (**Li et al., 1997**; **Loosli et al., 1999**; **Zuber et al., 2003**). The first morphogenetic sign of retinal development is the formation of the optic vesicle (OV) that evaginates from the wall of the developing diencephalon (**Figure 1a**, 0–1 day post-fertilization [dpf]). Although the molecular mechanism of OV formation is not completely understood, studies performed in different vertebrate models indicate that the retinal-specific homeodomain transcription factor Rx is involved in this process (**Loosli et al., 2003**; **Mathers et al., 1997**; **Medina-Martinez et al., 2009**; **Rembold et al., 2006**; **Stigloher et al., 2006**). As one of the earliest genes indicative for the retinal lineage, *Rx* is expressed in the anterior neural plate and later in the neuroepithelium of the OV. In *Rx3*-null mutants of several vertebrate species, the OV fails to evaginate (**Bailey et al., 2004**; **Loosli et al., 2003**; **Mathers et al., 1997**; **Voronina, 2003**). Additionally, *Rx3*-deficient cells are excluded from OV domains in fish (**Loosli et al., 2003**) and embryonic mouse chimeras (**Medina-Martinez et al., 2009**), demonstrating the crucial role of Rx3 in early retina development and morphogenesis.

Soon after evagination, two major domains are being specified within the OV (**Figure 1a**, 2 dpf). The distal/ventral region expresses retina-specific genes, marking prospective retinal territory, while the dorsal/outer region expresses the transcription factor Otx2, marking prospective retinal

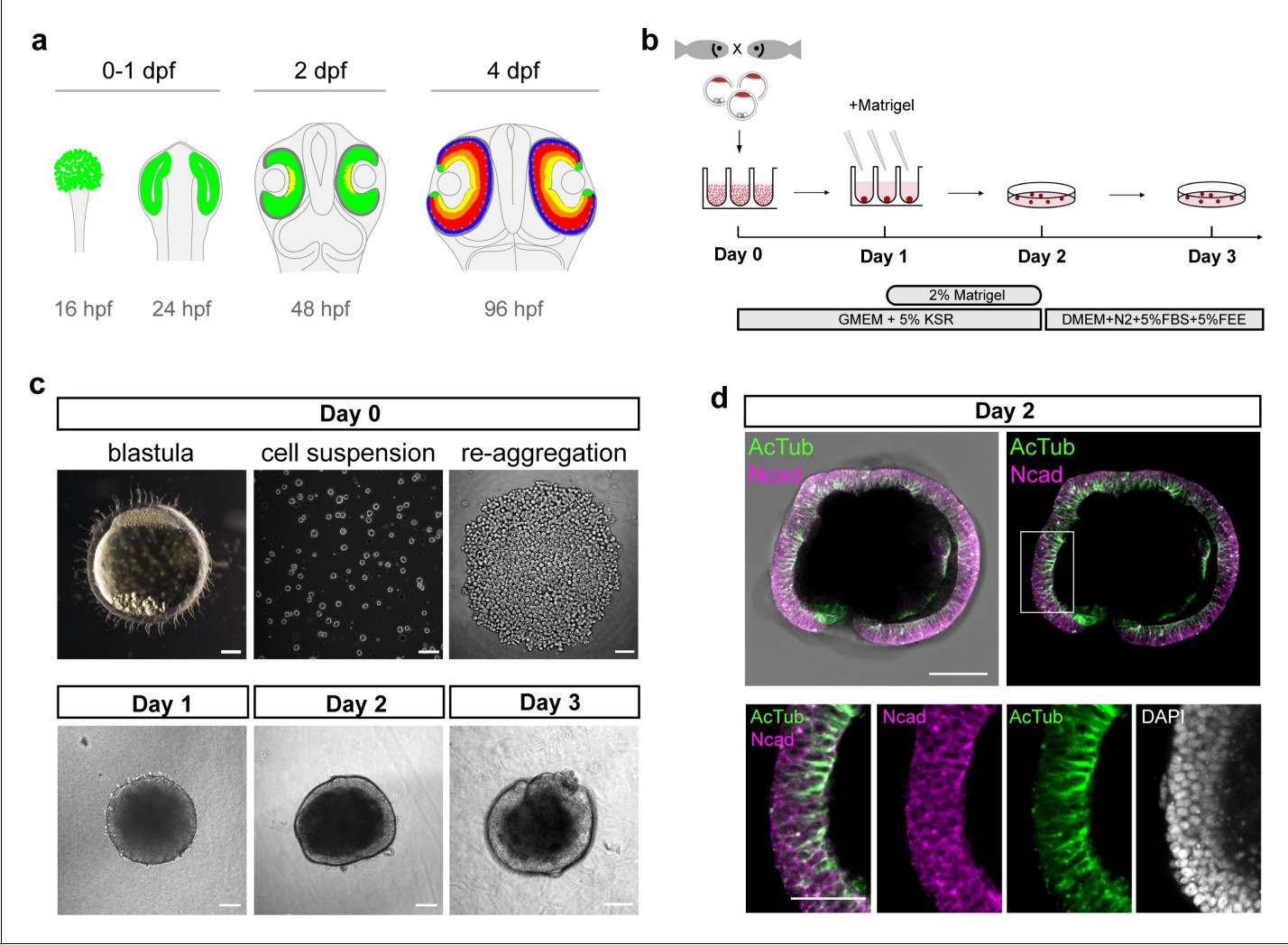

**Figure 1.** Generation of medaka fish primary pluripotent cell-derived aggregates. (**a**) Scheme representing stages and timing of the medaka fish retinal development. The retinal domain is indicated in green. Establishment of the eye field within the anterior neural plate is followed by the formation of optic vesicles at 1 day post-fertilization (dpf). Optic vesicle evagination is followed by the morphogenesis of a bi-layered optic cup formed by retina surrounded by retinal pigmented epithelium (RPE) and subsequent onset of retinal differentiation at 2 dpf. By 4 dpf, the major retinal cell types – retinal ganglion cells (yellow), amacrine cells (orange), bipolar cells (red), horizontal cells (cyan), and photoreceptor cells (blue) – are generated. (**b**) Schematic representation of aggregate generation, its timeline and culture conditions. At day 0, primary pluripotent cells were harvested from blastula-stage medaka embryos and re-aggregated in low binding U-shape 96-well plates. At day 1, the culture media was supplemented with Matrigel. At day 2, the aggregates were transferred to a low binding culture plate and maintained in 3D suspension culture conditions in DMEM/F12 media supplemented with 5% FBS, 5% FEE, and N2 supplement. The gross morphology of the aggregates was analyzed at days 1, 2, and 3. KSR, knockout serum replacement; FBS, fetal bovine serum; FEE, fish embryonic extract. (**c**) Dark-field images of a blastula-stage embryo, a blastula-derived cell suspension, and re-aggregated cells and the gross morphology of aggregates at days 1, 2, and 3 after re-aggregation. (**d**) Optical section showing aggregate organization at day 2 visualized by immunostaining against neuroepithelium-specific markers, N-cadherin (Ncad), and acetylated tubulin (AcTub), co-stained with DAPI nuclear stain. Scale bars: 100 and 50 µm (enlargement in (d)).

The online version of this article includes the following figure supplement(s) for figure 1:

**Figure supplement 1.** Impact of time point of medaka cell extraction on aggregation efficiency.

**Figure supplement 2.** Interior composition of day 2 organoids.

pigmented epithelium (RPE) territory (*Bovolenta et al., 1997*; *Fuhrmann, 2010*; *Hatakeyama et al., 2001*; *Hirashima et al., 2008*). Following a precisely coordinated morphogenesis (*Heermann et al., 2015*), the OV further forms the two-layered optic cup: the outer layer giving rise to the RPE and the inner layer forming retina populated with mitotically active retinal progenitor cells (reviewed by *Casey et al., 2021*; *Chow and Lang, 2001*; *Fuhrmann, 2010*). Soon after optic cup formation,

retinal progenitor cells start to differentiate into seven retinal cell types that together form the structure of the adult retina: retinal ganglion cells, amacrine cells, bipolar cells, Müller glia cells, horizontal cells, and (rod, cone) photoreceptors (*Young, 1985*; *Figure 1a*, 4 dpf).

Although many aspects of retinal development are known, its mechanistic analysis in vivo is challenging due to the complex environment of an embryo, including signals from the surrounding tissues. Taking advantage of cell assembly into various organ-like structures, in vitro systems provide a unique opportunity to study the basic principles of organ formation. Already in the 1940s, experiments in amphibians showed that early embryonic tissues can assemble to form neuronal tissues when cultured under specific conditions (*Simian and Bissell, 2017*). Animal cap ectoderm of *Ambystoma maculatum* salamander embryos was shown to differentiate into forebrain and occasionally retinal tissues in the absence of inductive signals, when cultured in a simple saline solution (*Barth, 1941*; *Holtfreter, 1944*; *Hurtado and De Robertis, 2007*). More recently, mouse and human embryonic stem (ES) cells have been shown to form retinal tissue when aggregated and cultured under 3D suspension culture conditions (*Eiraku et al., 2011*; *Kuwahara et al., 2015*; *Nakano et al., 2012*). However, so far, the organoid field has been restricted to mammalian species, leaving the ability of ES cells derived from different species to assemble in retinal tissue unresolved.

ES cells derived from teleosts (medaka and zebrafish) blastula-stage embryos are able to contribute to all germ layers of chimeric embryos, indicating that the ES cell pluripotency is conserved across vertebrate species (*Ho et al., 2014*; *Hong et al., 1998*; *Hong et al., 1996*; *Peng et al., 2019*; *Robles et al., 2011*; *Yi et al., 2010*). It has recently been shown that zebrafish blastula explants can form polarized structures that recapitulate aspects of embryonic patterning and morphogenesis when cultured in the absence of yolk (*Fulton et al., 2020*; *Schauer et al., 2020*). However, under no condition reported so far, fish-derived explants or re-aggregates have formed highly organized neural structures.

Here, we demonstrate that primary embryonic pluripotent cells derived from medaka and zebrafish form aggregates of neuroepithelial identity. These aggregates adopt retinal fate following the species-specific pace of development. Strikingly, the addition of extracellular matrix components (Matrigel) allows those to form retinal organoids with multiple cell layers. By adjusting the cell-seeding density, we directed organoids to form multiple OVs and non-retinal domains mimicking architecture of a 'head with two eyes'. Based on 3D in vivo imaging, we show that cell migration driving OV morphogenesis in these organoids is intriguingly reminiscent of the in vivo situation. Additionally, fish-derived retinal organoids retain the genetic constraints and the developmental program displayed in the embryo. Altogether, within only 4 days of culture, fish-derived primary embryonic pluripotent cells efficiently proceed through retinal differentiation, OV morphogenesis, and the onset of retinal differentiation, offering new ways to address multiple aspects of development and to systematically probe the dependence of organogenesis on variable physical environments.

## Results

### Generation of fish primary embryonic pluripotent cell-derived aggregates

Here, we used primary embryonic pluripotent cells derived from blastula-stage embryos of medaka (*Oryzias latipes*) as a source of pluripotent cells and established the conditions to generate the anterior neural structures, particularly retinal tissue. Previous studies performed with mouse and human ES cell aggregates have shown that low serum concentration in combination with 3D suspension culture support retinal

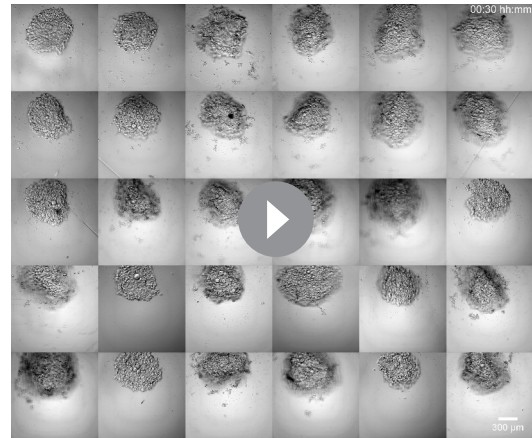

**Video 1.** Aggregation and compaction of blastula-derived cells. Time-lapse imaging of medaka-derived blastula cells going through the process of aggregation and compaction. Imaging was performed with 30 min intervals on all aggregates (n = 30) in the batch. Scale bar: 100 μm.
https://elifesciences.org/articles/66998#video1

specification and differentiation. In particular, the addition of extracellular matrix components such as laminin-rich Matrigel promotes retinal formation (*Eiraku et al., 2011*; *Nakano et al., 2012*). We dissociated blastula-stage embryos and used U-shaped low adhesive wells to re-aggregate 1000–2000 cells in basic media supplemented with 5% KSR (knockout serum replacement) (*Figure 1b,c*). The aggregation and compaction of cells was highly efficient (100%; n = ~1500 from 54 independent experiments) and resulted in a smooth and compacted morphology by day 1 (*Figure 1c*; *Video 1*). The potential of blastomeres to form aggregates was restricted to the blastula stage (1000–2000 cells) and cells derived from earlier developmental stages such as early and late morula failed to form stable aggregates or aggregated only partially (*Figure 1—figure supplement 1*). Thus, the exact time point of cell dissociation plays a decisive role for the success and efficiency of aggregation.

After 1 day of culture, media were supplemented with 2% Matrigel (*Figure 1b*). The aggregates then formed an organized and layered epithelium surrounding the aggregate from day 2 onward (*Figure 1c*) and the cells of the peripheral layer expressed neuroepithelial markers, for example, N-cadherin and acetylated tubulin, indicating that cells adopted neuroepithelial identity (*Figure 1d*, *Figure 1—figure supplement 2*). 3D imaging of aggregates at day 2 revealed a complex internal morphology, consisting of lumens and cells of neuroepithelial identity (*Figure 1—figure supplement 2*). Based on the higher order epithelial organization, we call the aggregates from day 2 on organoids.

## Fish-derived organoids form retinal neuroepithelium under the control of *Rx3*

Retinal specification in vertebrates is governed by the action of retina-specific transcription factors such as Rx, Pax6, Six3, Sox2, Six6, and Lhx2 within the anterior neural plate (*Li et al., 1997*; *Loosli et al., 1999*; *Zuber et al., 2003*) and the formation of the OV neuroepithelium (*Chow and Lang, 2001*; *Fuhrmann, 2010*; *Martinez-Morales et al., 2017*). *Rx* genes, in teleosts represented by three paralogous genes (*Rx1*, *Rx2,* and *Rx3*), are the earliest genes expressed by the retinal lineage (*Chuang et al., 1999*; *Deschet et al., 1999*; *Loosli et al., 2003*; *Loosli et al., 2001*). To analyze the efficiency of retinal fate acquisition, we monitored the earliest expressed *Rx* paralogue, *Rx3*, and employed a transgenic reporter line (*Rx3::H2B-GFP*) in which nuclear GFP is expressed under the control of the *Rx3* regulatory elements (*Figure 2a,b*; *Rembold et al., 2006*). *Rx3::H2B-GFP* drives the expression of H2B-GFP already in the anterior neural plate and is subsequently found in the neuroepithelium of the OV and prospective forebrain of fish embryos 1 dpf (*Figure 2a*). We addressed the onset of GFP expression in *Rx3::H2B-GFP* aggregates by time-lapse imaging over a time span of 18 hr post-aggregation (hpa) and compared it to corresponding *Rx3::H2B-GFP* embryos for reference (*Figure 2c,d*; *Video 2*). All aggregates started expressing GFP at 15.5 ± 0.36 hpa (n = 17) with GFP expression in control *Rx3::H2B-GFP* embryos starting at 13.5 ± 1.03 hr (n = 6) after the blastula stage (*Figure 2c*), corresponding to the onset of *Rx3* expression at around 20 hrs post-fertilization (hpf) (*Loosli et al., 2001*). In comparison to the embryo, the organoids showed a delay of GFP expression of approximately 2–3 hr which corresponds to the time of re-aggregation (*Video 1*). This data shows that following our protocol, the re-aggregation of primary embryonic pluripotent cells results in an efficient acquisition of retinal fate.

To address whether cell aggregation is a prerequisite for retinal specification or, alternatively, cells can acquire retinal fate autonomously, we followed the onset of *Rx3* expression in individual blastula-stage cells. *Rx3::H2B-GFP*-derived primary embryonic pluripotent cells were cultured under suspension culture condition, that is, gentle rocking to prevent cell aggregation and analyzed for GFP expression 24 hr later (*Figure 2—figure supplement 1a,b*). Under those conditions, cells formed GFP-expressing clusters, indicating that cells acquired retinal fate even without aggregation. To further refine the analysis and address whether isolated single primary embryonic pluripotent cells give rise to GFP-expressing clusters, cells were cultured individually in the 96-well plate and imaged over time (*Figure 2—figure supplement 1a,c*; *Video 3*).

Also under those conditions, individual cells gave rise to GFP-expressing clones, showing that cells acquire retinal fate autonomously. Similar to GFP expression in aggregated primary embryonic pluripotent cells (*Figure 2c,d*; *Video 2*), single cell-derived clones started to express GFP at about 16 hr (*Video 3*), indicating that the onset of retinal fate is genetically timed and independent of cellular environment.

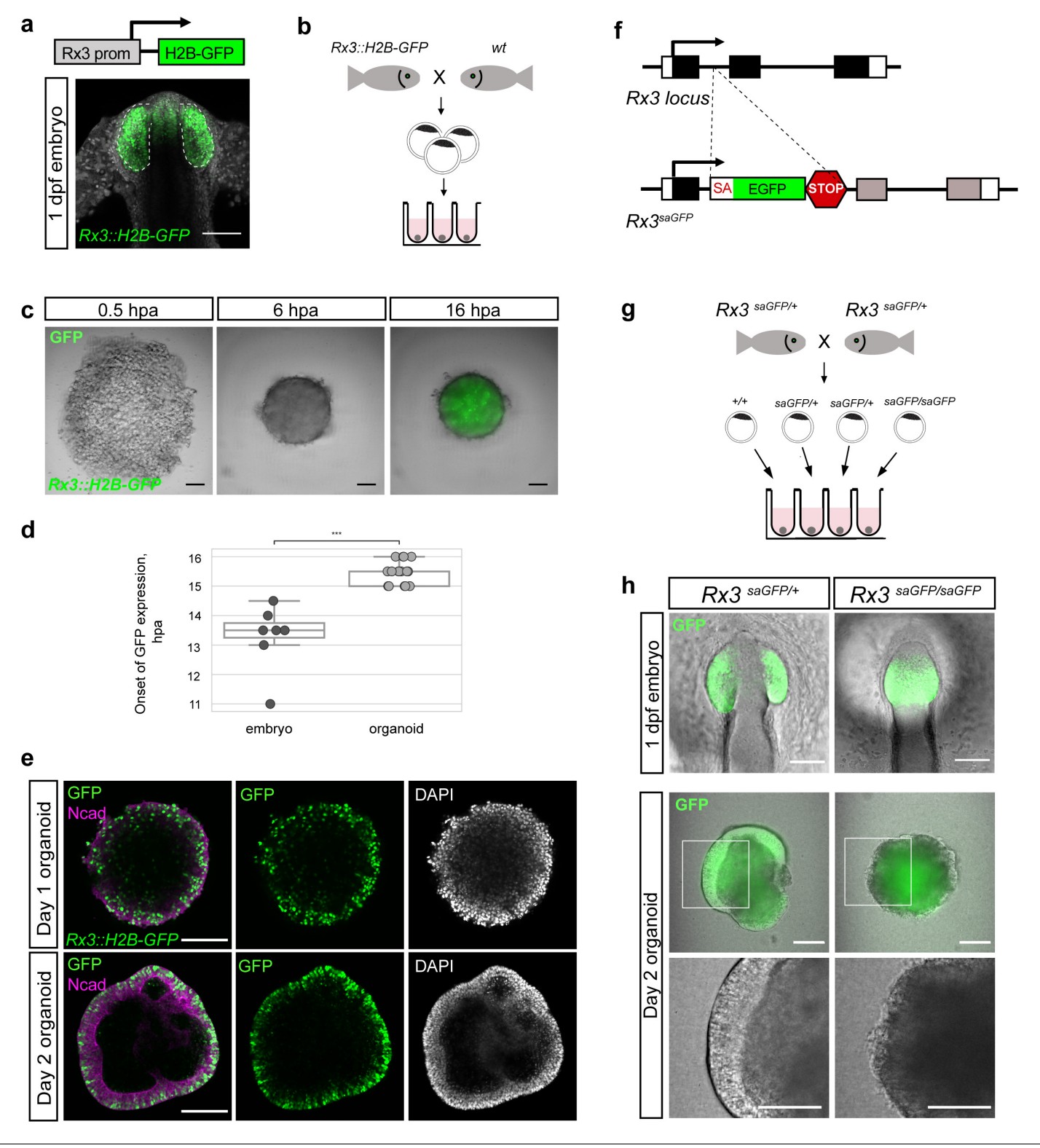

**Figure 2.** Medaka-derived organoids form retinal neuroepithelium under the control of *Rx3*. (a) Schematic representation of the *Rx3::H2B-GFP* transgenic construct and the corresponding expression domain of GFP in the optic vesicles of a developing medaka embryo at 1 dpf. (b) Scheme of organoid generation from *Rx3::H2B-GFP* transgenic fish. (c) Bright-field and fluorescence images of aggregates derived from *Rx3::H2B-GFP* transgenic fish at 0.5, 6, and 16 hpa. (d) Analysis of the onset of GFP expression in *Rx3::H2B-GFP*-derived fish embryos (n = 6) and organoids (n = 17). ***p<0.001. (e) Optical sections of day 1 (before the addition of Matrigel) and day 2 organoids derived from *Rx3::H2B-GFP* transgenic fish stained with antibodies

*Figure 2 continued on next page*

*Figure 2 continued*

against N-cadherin (Ncad) and GFP, co-stained with DAPI nuclear stain. (**f**) Generation of *Rx3KO* (*Rx3^saGFP^*) line – schematic representation of the *Rx3* locus with integrated saGFP-OPT cassette. An open reading frame-adjusted gene trap cassette comprising a splice acceptor and a *GFP* sequence (saGFP) followed by a polyA and a strong terminator sequence derived from the ocean pout (OPT; STOP) were inserted into the *Rx3* locus. (**g**) Scheme of organoid generation from *Rx3*-deficient single blastulae. (**h**) Bright-field and fluorescence images of phenotypes of *Rx3^saGFP/+^*(*Rx3* +/- heterozygote) and *Rx3^saGFP/saGFP^* (*Rx3* -/- homozygote) mutants at 1 dpf and corresponding organoids at day 2. dpf, days post-fertilization; hpa, hours post-aggregation; hpf, hours post-fertilization; wt, wild type. Scale bar: 100 µm.

The online version of this article includes the following figure supplement(s) for figure 2:

**Figure supplement 1.** Medaka primary embryonic pluripotent cells acquire retinal fate cell autonomously, independently of cell aggregation.

**Figure supplement 2.** Impact of Matrigel addition on neuroepithelium organization.

**Figure supplement 3.** Expression of retina-specific factors Rx2, Sox2, and Lhx2 in retinal organoid neuroepithelium.

**Figure supplement 4.** Characterization of *Rx3^saGFP^* line.

Interestingly, *Rx3* expression was initiated also in the absence of Matrigel already at day 0, indicating that the acquisition of retinal fate occurs spontaneously and independently of extracellular matrix components. Further analysis showed that GFP-expressing cells are localized in disperse manner before the addition of Matrigel, that is, on day 1 (*Figure 2e*). Matrigel addition triggered the formation of a neuroepithelium by day 2 and confined GFP-expressing cells to the peripheral region of the organoid. The formation of a neuroepithelium was clearly dependent on the action of Matrigel (*Figure 2—figure supplement 2*).

To address whether GFP-positive aggregates followed retinal specification, we analyzed the expression of retinal markers expressed subsequently during retinal specification in vivo such as Rx2, Sox2, and Lhx2. The expression of these transcription factors in the epithelium of day 2 organoids resembled their expression in the neuroepithelium of OV in the 1 dpf embryo (*Figure 2—figure supplement 3*), indicating that the aggregates adopted the complex retinal fate. All organoids (n = ~1500) from 52 independent experiments formed retinal neuroepithelia, indicating that retinal specification under the established conditions is robust and highly efficient.

The formation of the evaginated OV strictly depends on expression of the transcription factor Rx such that in *Rx3*-null mutants, OVs fail to form leading to an eyeless phenotype (*Loosli et al., 2003*; *Loosli et al., 2001*; *Mathers et al., 1997*; *Rembold et al., 2006*; *Stigloher et al., 2006*). Organoids derived from *Rx3*-null mutants can be used to investigate the dependence of the formation of the retinal neuroepithelium on Rx3. Unfortunately, null alleles resulting from mutagenesis screens do not contain traceable markers. We therefore followed a Crispr/Cas targeted gene-trapping approach into an intron of *Rx3*. We inserted an open reading frame-adjusted gene trap cassette comprising a splice acceptor and a *GFP* sequence (saGFP) for visual readout followed by a polyA and strong terminator sequence derived from the ocean pout (OPT; *Clark et al., 2011*, *Figure 2—figure supplement 4*). Using the CRISPR/Cas9 system for targeted integration, we generated traceable *Rx3* mutants (*Rx3^saGFP^*; *Figure 2—figure supplement 4*). While heterozygotes *Rx3^saGFP/+^* develop normally and give rise to GFP labeled OVs, homozygous-mutant embryos (*Rx3^saGFP/saGFP^*) fail to evaginate the OVs from the lateral wall of the diencephalon (*Figure 2g*, *Figure 2—figure supplement 4*). In contrast to the *eyeless* mutant described (*Loosli et al., 2001*), the mutant phenotype, that is the inability to evaginate the OV, was not temperature sensitive. To address whether fish-derived organoids retain the genetic constraints and the developmental program displayed in the embryo, we derived organoids from individual *Rx3^saGFP/saGFP^* and *Rx3^saGFP/+^* blastulae (*Figure 2g*). While a retinal neuroepithelium formed in *Rx3^saGFP/+^* (n = 9) organoids by day 2, aggregates derived from individual *Rx3*-mutant homozygote blastulae (*Rx3^saGFP/saGFP^*) (n = 4) did not form a neuroepithelium and aggregates were lacking regular morphology by day 2 (*Figure 2g*) and thus resembled organoids cultured in the absence of Matrigel. These data indicate that the formation of retinal neuroepithelium is hardwired also in the organoid and follows the same developmental program as in vivo, also upon loss-of-function (*Loosli et al., 2001*; *Rembold et al., 2006*; *Stigloher et al., 2006*; *Winkler et al., 2000*).

We next asked whether the spontaneous acquisition of retinal fate is generally applicable to other fish species, for example, the evolutionarily distant zebrafish (*Danio rerio*). We used zebrafish blastulae as the source for primary embryonic pluripotent cells to establish zebrafish retinal organoids. To monitor the acquisition of retinal fate, we used the early retinal marker gene *Rx3* as detailed above.

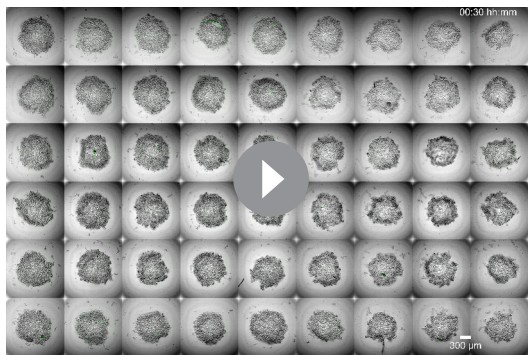

**Video 2.** Acquisition of retinal fate within medaka blastula-derived aggregates. Time-lapse imaging of *Rx3::H2B-GFP* medaka-derived blastula cells going through the process of aggregation, compaction, and acquisition of retinal fate (GFP expression). Imaging was performed with 30 min intervals on all aggregates (n = 54). Scale bar: 100 μm.
https://elifesciences.org/articles/66998#video2

We used a transient assay and injected a *Rx3::H2B-GFP* reporter construct into one cell stage embryos and generated cell aggregates when embryos reached blastula stage (*Figure 3a*).

Zebrafish cells aggregated as efficiently as medaka cells and displayed an epithelial organization already by day 1 (*Figure 3b*). Considering the faster early development of zebrafish compared to medaka (*Iwamatsu, 2004*; *Kimmel et al., 1995*), we asked whether the difference in embryonic development impacts on the re-aggregation and subsequent differentiation of aggregates. We performed time-lapse imaging in zebrafish aggregates over 18 hpa and addressed the onset of retinal differentiation by *Rx3*-driven GFP expression. Aggregates generated from zebrafish embryos formed organoids that started expressing GFP at 9.75 ± 0.21 hpa (n = 15). As in medaka, the onset of *Rx3* expression in the forming organoids was delayed by 2–3 hr in comparison to the corresponding *Rx3::H2B-GFP* reporter embryos (*Figure 3c*; *Video 4*) that started to express GFP at 7.88 ± 0.77 hr (n = 6) after the blastula stage. The cells followed their endogenous program consistently and were only delayed by the dissociation/aggregation procedure. Thus, the formation of a neuroepithelium in aggregates reflects the species-specific difference in the relative pace of development.

*Rx3* reporters for retina formation (*Rx3::H2B-GFP*) displayed H2B-GFP expression in the developing retina at 1 dpf and aggregates formed organoids showing retinal identity as indicated by *Rx3* expression in the outer cellular layer (*Figure 3d*) by day 1. This data shows that under the established conditions, fish primary embryonic pluripotent cells are guided to differentiate into a retinal neuroepithelium – a feature found over a wide evolutionary distant medaka and zebrafish, indicating a deep conservation of early retinal development irrespective of the embryonic environment.

## Fish organoids form OV-like structures

Since primary embryonic pluripotent cells from both, medaka and zebrafish, efficiently form retinal organoids, we subsequently focused our experiments on one organism – medaka.

Development of a functional retina is accompanied by several morphological transitions that ultimately result in the formation of a functional retina. One of them is the formation of the OV. When organoids were generated by the aggregation of 1000–2000 cells (referred to as >1000 cells), the approximate number of cells in a single blastula, OV formation was not favored (*Figure 4a*) and instead these organoids formed a continuous neuroepithelium on their entire surface. This neuroepithelium expressed retina-specific markers (*Rx2* and *Rx3::H2B-GFP*) all around the surface of organoid (*Figure 4a–b*) indicating the formation of a single retinal anlage. It has

00:30 hpa

**Video 3.** Acquisition of retinal fate within individual medaka – derived primary embryonic pluripotent cells. Time-lapse imaging of medaka-derived blastula cells derived from *Rx3::H2B-GFP* transgenic fish. Imaging was performed with 30 min intervals. Scale bar: 100 μm.
https://elifesciences.org/articles/66998#video3

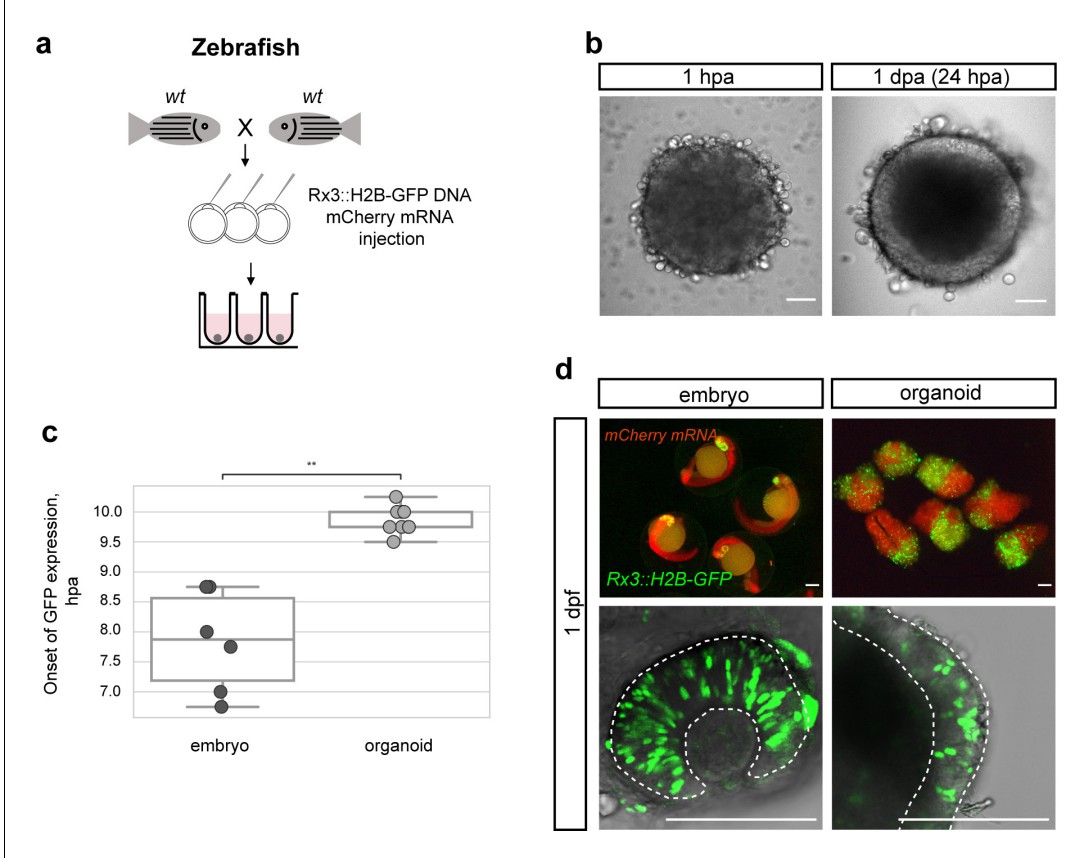

**Figure 3.** Zebrafish blastula-derived primary embryonic pluripotent cells forming retinal organoids. (a) Scheme of aggregate generation from *Rx3::H2B-GFP* DNA-injected blastulae. (b) Bright-field images of zebrafish-derived aggregates at 1 hpa and 1 dpa. (c) Analysis of the onset of GFP expression in *Rx3::H2B-GFP*-derived zebrafish embryos (n = 6) and organoids (n = 15). **p<0.01. (d) Fluorescent images of mCherry and GFP expression domains in *Rx3::H2B-GFP*-injected embryos at 1 dpf and corresponding aggregates at day 1. Dashed lines indicate the embryonic retina and organoid retinal neuroepithelium, respectively. wt, wild type; dpf, days post-fertilization; dpa, days post-aggregation; hpa, hours post-aggregation. Scale bar: 100 μm.

been previously reported that in some in vitro self-organizing systems, the number of interacting cells can generate a bias toward particular morphological processes (*van den Brink et al., 2014*; *Fulton et al., 2020*; *Völkner et al., 2016*). Thus, we reduced the cell-seeding density by 50%. Interestingly, those organoids ('small organoids', generated by the aggregation of <1000 cells) reproducibly displayed expression of Rx2 and *Rx3* only in restricted regions (*Figure 4a,b*) reminiscent of the forming OV in vivo. While organoids generated by aggregation of >1000 cells established a single, uniform, domain of retinal neuroepithelium (n = 35/35), small organoids displayed a more complex and variable morphology, forming one to four isolated OV-like retinal domains (n = 92/123 one domain, n = 27/123 two domains, n = 3/123 three domains, and n = 1/123 four domains) (*Figure 4c*, *Figure 4—figure supplements 1* and *2*). The size of these isolated retinal domains in small organoids (154.5 ± 28.6 μm; n = 56) was reminiscent of the OVs in embryos at 1 dpf (162 ± 10.6 μm; n = 16) (*Figure 4d*).

Interestingly, retinal domains formed in small organoids displayed OV-like morphology with specifically localized expression of Rx2 and Otx2 transcription factors, marking prospective retinal and RPE territories respectively (*Figure 4e*) and indicating further compartmentalization of the organoid-derived OV.

Besides the retinal tissue, small aggregates also established an Rx2-negative, non-retinal domain, expressing the general neuronal marker Sox2 (*Pevny and Placzek, 2005*; *Wegner and Stolt, 2005*) and the forebrain marker Otx2 (*Acampora et al., 1995*; *Simeone et al., 1993*; *Tian et al., 2002*), strongly indicating that the neighboring tissue adopted anterior neuronal identity (*Figure 4f*). In a fraction (22%) of organoids, intriguingly, retinal and non-retinal domains showed an arrangement,

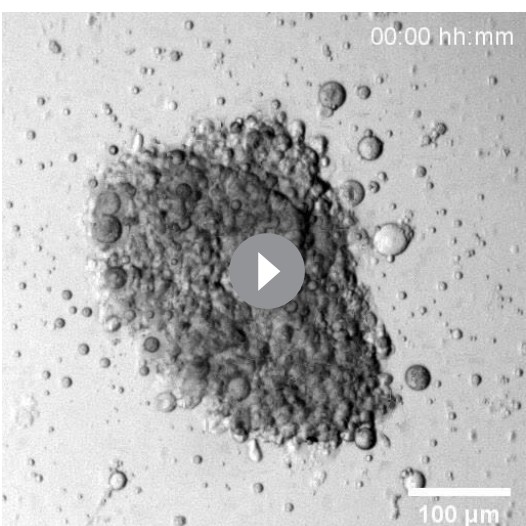

**Video 4.** Acquisition of retinal fate within zebrafish blastula-derived aggregates. Time-lapse imaging of *Rx3::H2B-GFP* zebrafish-derived blastula cells going through the process of aggregation, compaction, and acquisition of retinal fate (GFP expression). Imaging was performed with 30 min intervals. Scale bar: 100 µm.
https://elifesciences.org/articles/66998#video4

resembling a 'head with two eyes' (n = 27/123) (*Figure 4e–f*, *Figure 4—figure supplements 1* and *2*).

To address the mode of OV formation in fish organoids, we employed live imaging of *Rx3:: H2B-GFP*-derived organoids from day 1 to day 2 of development. *Figure 5a* displays the development of the organoid forming two opposing OV structures. At the initial time point (24 hpa), GFP-positive cells were distributed throughout the whole volume of the organoid indicating that cells acquired retinal identity individually. Single cell tracking and analysis of cell displacement showed that initially cells behaved as freely diffusing particles (*Figure 5b*). This behavior changed at about 34 hpa, when GFP-positive cells showed directed migration – non-linear growth of mean squared displacement – with an estimated speed of 0.413–0.479 µm/min resulting in the morphogenesis of the OV-like structures by 40 hpa (*Figure 5a,b*; *Video 5*). A similar transition from linear to nonlinear diffusion, due to increasing density of nuclear packing, was recently demonstrated in vivo in the growing retina of 24 hpf zebrafish (*Azizi et al., 2020*). Directionality analysis of single cell tracks (*Figure 5c*) showed that 51% of the *Rx3*-expressing cells were moving from the center to the periphery (outward) of the organoid and contributed to the formation of the OV (see dark blue, red, orange, and yellow tracks in *Figure 5a*). About 49% of the initially laterally located cells moved in the opposite direction – toward the interior of the organoid (inward) (light green track in *Figure 5a*). These outward- and inward-directed cell movements were highly reminiscent of the migratory behavior of *Rx3*-expressing cells during OV formation in medaka fish (*Rembold et al., 2006*). Notably, a similar behavior was observed in the organoids forming single and multiple OVs (*Video 6*). Altogether these results show that the cell behavior in fish organoids closely recapitulates the corresponding behavior during development in vivo.

It is worth mentioning that the organization of retina-committed cells into the OV neuroepithelium is strictly dependent on the presence of extracellular matrix proteins as *Rx3*-expressing cells attempt but fail to form OV structures in the absence of Matrigel (*Video 7*). Consistently, laminin-1 (the main component of the Matrigel) is highly enriched specifically at the basal surface of the forming OV and has been reported to play an essential role in cell polarization and maintenance of neuroepithelial cell morphology during OV evagination (*Ivanovitch et al., 2013*).

## Fish organoids show onset of retinal differentiation

Retinal specification is followed by the process of differentiation which leads to the generation of seven retinal cell types (retinal ganglion cells, amacrine cells, horizontal cells, bipolar cells, rod and cone photoreceptors, and Müller glia cells) organized in three nuclear layers. One of the first hallmarks of retinal differentiation is the expression of the transcription factor Atoh7 (*Brown et al., 1998*; *Del Bene et al., 2007*; *Kay et al., 2001*). Atoh7-positive progenitors have been found to give rise to retinal ganglion cells, amacrine cells, horizontal and photoreceptor cells during fish retinal development (*Poggi et al., 2005*). We monitored *Atoh7* expression in medaka-derived retinal organoids, using an *Atoh7::EGFP* transgenic line (*Del Bene et al., 2007*; *Figure 6a–c*). In embryos, EGFP expression was located specifically in the differentiating retina at 2 dpf (*Figure 6c*). Day 3 organoids generated from *Atoh7::GFP* blastulae (generated by the aggregation of <1000 cells) showed specific morphology with EGFP-negative non-retinal and EGFP-expressing retinal domains (*Figure 6b*; *Video 8*), indicating that the retinal differentiation program was successfully initiated. Furthermore, non-retinal regions proceeded through the process of neuronal differentiation as indicated by

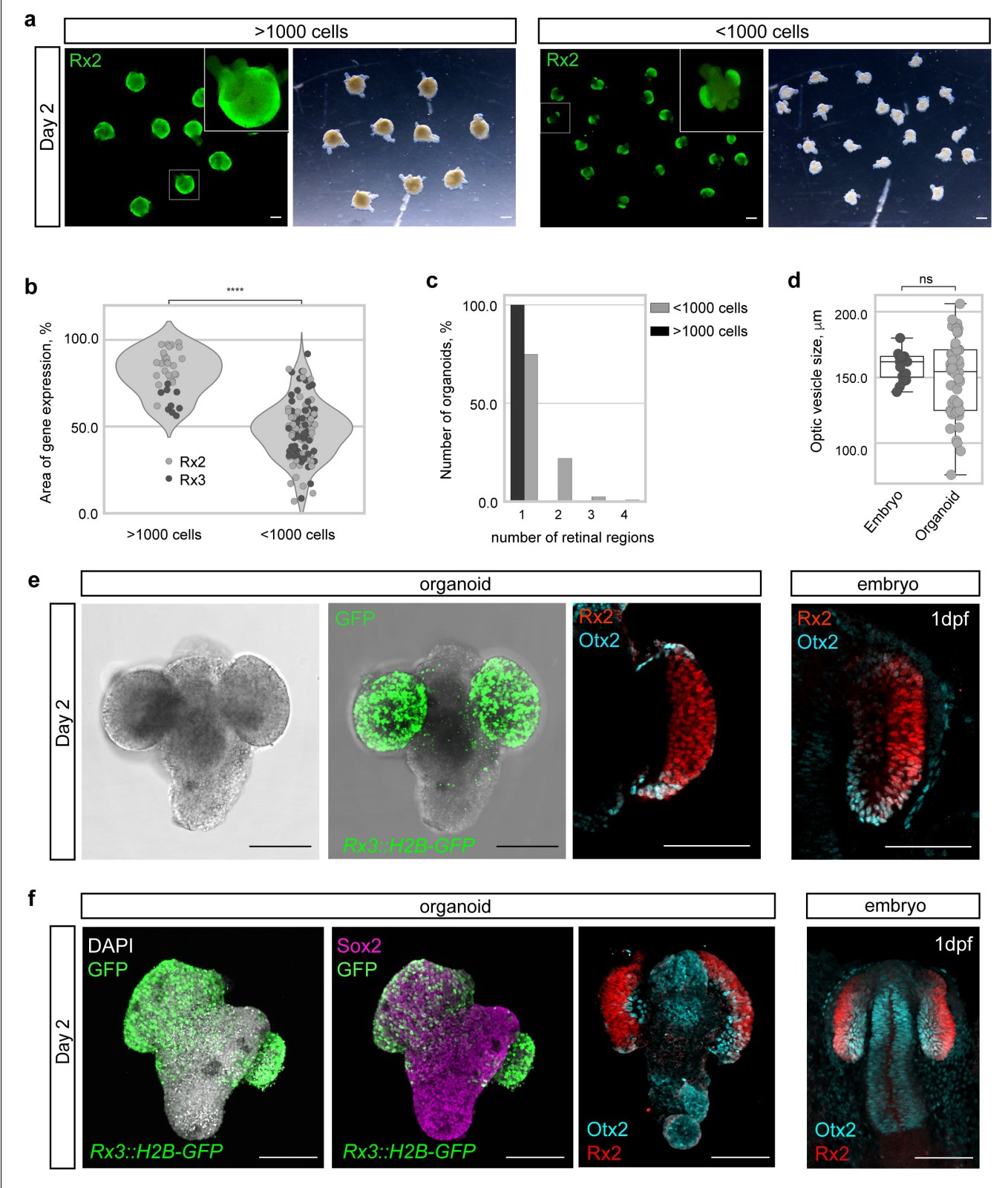

**Figure 4.** Medaka primary embryonic pluripotent cells form optic vesicle-like structures. (a) Fluorescence and bright-field images of day 2 organoids produced by aggregation of >1000 cells and <1000 cells stained with anti-Rx2 antibody. (b) Analysis of the area of Rx2 (wild-type organoids stained with anti-Rx2 antibody) and *Rx3* (*Rx3::H2B-GFP*-derived organoids stained with anti-GFP antibody) expression area (% of total organoid area) in day 2 organoids. ****p<0.0001. (c) Number of organoids forming (1–4) individual retinal regions produced by aggregation of >1000 (n = 26 for Rx2, n = 9 for

*Figure 4 continued on next page*

*Figure 4 continued*

*Rx3*) and <1000 (n = 57 for Rx2, n = 66 for *Rx3*) cells from nine independent experiments. (d) Size, measured as largest circumference, of the optic vesicle of 1 dpf embryos and optic vesicle-like structures formed by day 2 organoids (n = 16 embryos, n = 56 aggregates from six independent experiments). ns, non-significant. (e) Bright-field and fluorescent images of day 2 *Rx3::H2B-GFP* organoids stained with anti-GFP antibody. Optical sections of an organoid (day 2) (n = 9/10) and an embryo (1 dpf) stained with anti-Rx2 and anti-Otx2 antibodies. (f) Maximal projection of day 2 organoids and 1 dpf embryo generated from *Rx3::H2B-GFP* transgenic or wild-type blastulae and stained with neural tissue-specific anti-Sox2 (n = 12/12) and anti-Otx2 (n = 10/10) antibodies, co-stained with anti-Rx2 and DAPI nuclear stain. hpa, hours post-aggregation; dpf, days post-fertilization. Scale bars: 100 μm.

The online version of this article includes the following figure supplement(s) for figure 4:

**Figure supplement 1.** Overview of complexity and variability of organoid morphology generated within one experiment.

**Figure supplement 2.** Examples of organoids generated by aggregation of <1000 cells forming one, two, and three optic vesicle-like structures.

expression of HuC/D (*Figure 6c*; *Video 8*), a marker of early post-mitotic neurons (*Good, 1995*; *Kim et al., 1996*; *Park et al., 2000*).

Furthermore, retinal domains of day 4 organoids consisted of differentiating retinal neurons organized in multiple layers of cells differentiating toward amacrine and ganglion (expressing HuC/D) (*Kay et al., 2001*; *Link et al., 2000*; *Park et al., 2000*), photoreceptor and bipolar (expressing Otx2) (*Fossat et al., 2007*; *Glubrecht et al., 2009*; *Koike et al., 2007*; *Nishida et al., 2003*), as well as horizontal (expressing Prox1) (*Dyer et al., 2003*) cell lineages (*Figure 6d*). Interestingly, the arrangement of the respective cellular layers was inverted when compared to normal retinal organization (*Figure 6e*). This can be attributed to the morphological differences, particularly the OV to optic cup transition, between retinal morphogenesis in vivo and organoid. In organoids, retinal differentiation

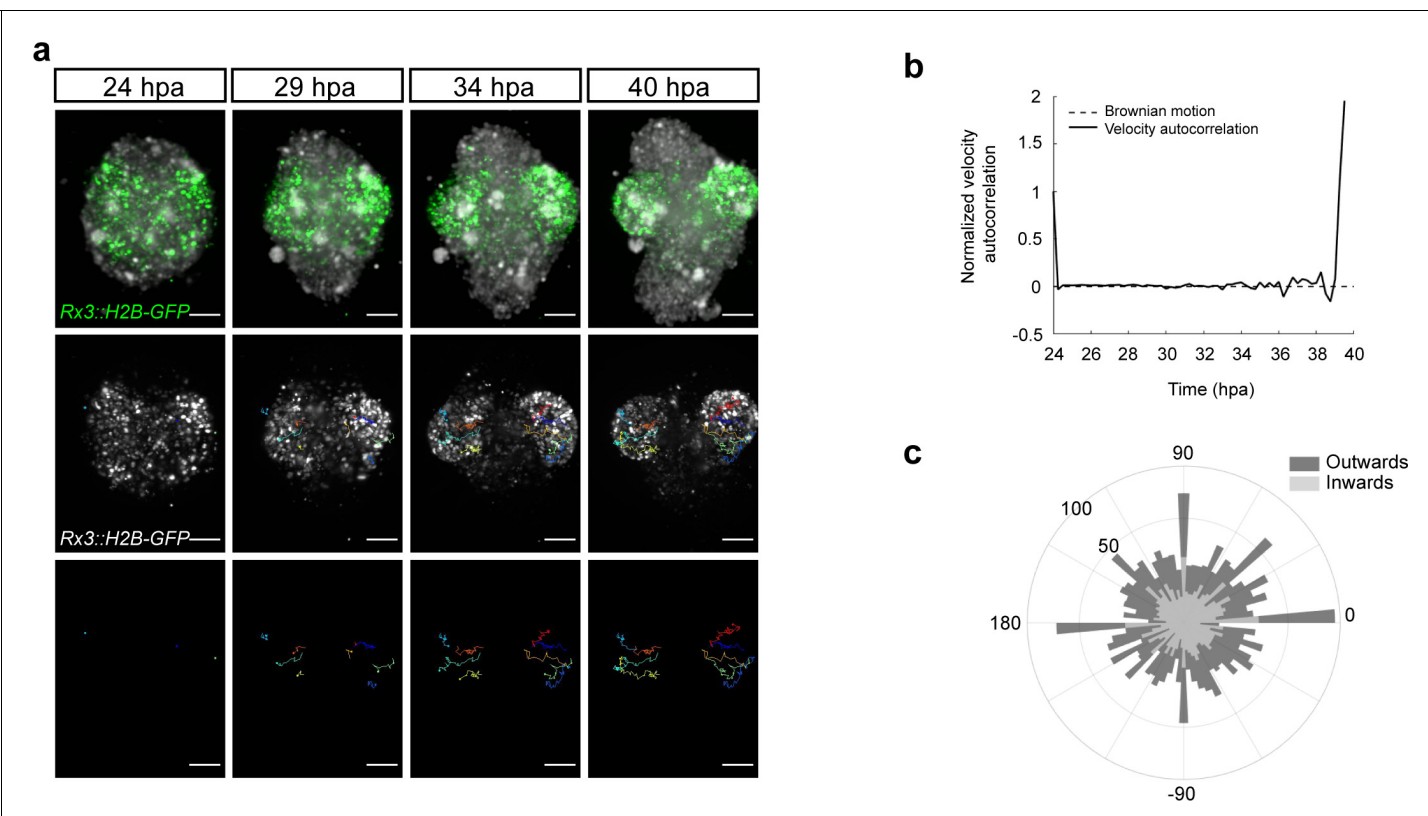

**Figure 5.** Dynamics of the optic vesicle formation in medaka organoids. (a) In vivo time-lapse images acquired with MuVi SPIM of optic vesicle-like structure evagination in *Rx3::H2B-GFP* retinal organoids. From top to bottom: maximal projection of quasi bright field, GFP, and tracks of exemplary cells. (b) Normalized velocity autocorrelation for all tracks (n = 4600; from two optic vesicles of one organoid). (c) Directional histogram of tracks (n = 4600; from two optic vesicles of one organoid) separated for inward and outward movements. hpa, hours post-aggregation; dpf, days post-fertilization. Scale bar: 100 μm.

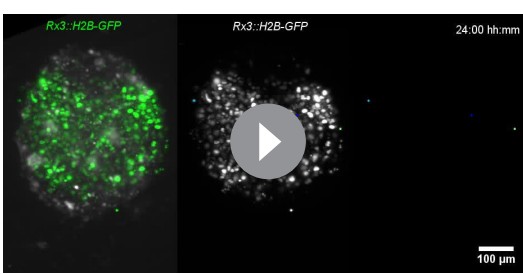

**Video 5.** Optic vesicle formation within blastula-derived retinal organoid. Time-lapse imaging of optic vesicle-like structure evagination in the medaka *Rx3::H2B-GFP* retinal organoid. Maximal projection of quasi bright field, GFP, and tracks of exemplary cells. Imaging was performed with 15 min intervals. Scale bar: 100 μm.
https://elifesciences.org/articles/66998#video5

is initiated without the optic cup formation, most probably leading to the inverted retinal organization.

Organoids generated by the aggregation of >1000 cells, which form retinal neuroepithelium around entire organoid surface, proceed through the process of retinal differentiation as well (*Figure 6—figure supplement 1*), suggesting that the OV formation is not a prerequisite for the retinal differentiation.

Although the organoid-derived OVs compartmentalize to prospective retinal and RPE domains by day 2 (*Figure 4e*), none of the organoids analyzed between day 3 and day 4 (n = 88, data not shown) showed significant level of pigmentation, the hallmark of RPE differentiation. This indicates that, similarly to human and mouse retinal organoid cultures (*Eiraku et al., 2011*; *Kuwahara et al., 2015*; *Nakano et al., 2012*), RPE fate was not favored under the established culture conditions. Previous studies showed that RPE fate determination in vivo, as well as in retinal organoids, is promoted by the activation of the Wnt/β-catenin signaling pathway (*Eiraku et al., 2011*; *Fujimura et al., 2009*; *Kuwahara et al., 2015*; *Nakano et al., 2012*; *Westenskow et al., 2009*). We thus promoted the Wnt/β-catenin pathway by treatment of the organoids from day 1 on with the GSK3β inhibitor CHIR-99021. Analysis at day 2 and day 4 indicated the presence of RPE (*Figure 6—figure supplement 2*). Compared to control (DMSO-treated) organoids, CHIR-99021-treated day 2 organoids showed a RPE-specific Otx2 expression on the expense of the retina-specific Rx2 expression (*Figure 6—figure supplement 2b*). By day 4, retinal domains in treated organoids (13/13) became pigmented with most cells expressing Otx2 (*Figure 6—figure supplement 2c*), demonstrating that those cells acquired RPE fate.

Taken together, our findings show that primary embryonic pluripotent cells derived from blastula-stage fish embryos form aggregates

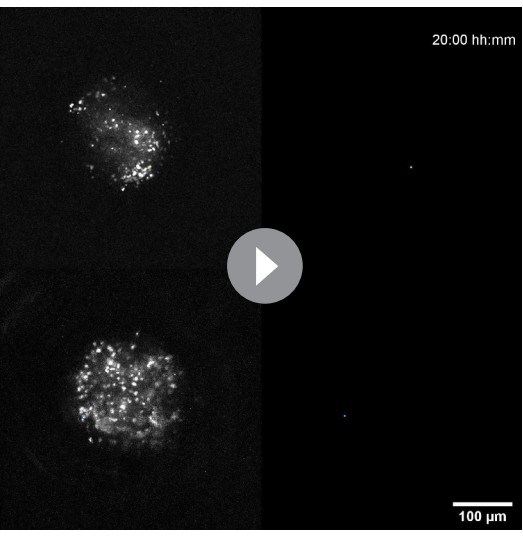

**Video 6.** Example of organoids forming single and multiple optic vesicle-like structures. Time-lapse imaging of optic vesicle-like structure evagination in medaka *Rx3::H2B-GFP* retinal organoids. Maximal projection of GFP and tracks of exemplary cells. Imaging was performed with 15 min intervals. Scale bar: 100 μm.
https://elifesciences.org/articles/66998#video6

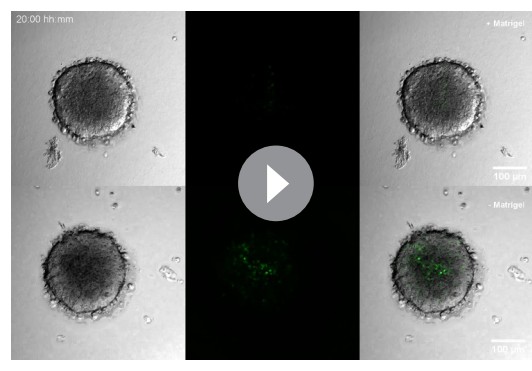

**Video 7.** Extracellular matrix controls formation of optic vesicle neuroepithelium. Time-lapse imaging of optic vesicle-like structure evagination in medaka *Rx3::H2B-GFP* retinal organoids in presence or absence of Matrigel (extracellular matrix component) shown as maximal projection of GFP and bright field. Imaging was performed with 30 min intervals. Scale bar: 100 μm.
https://elifesciences.org/articles/66998#video7

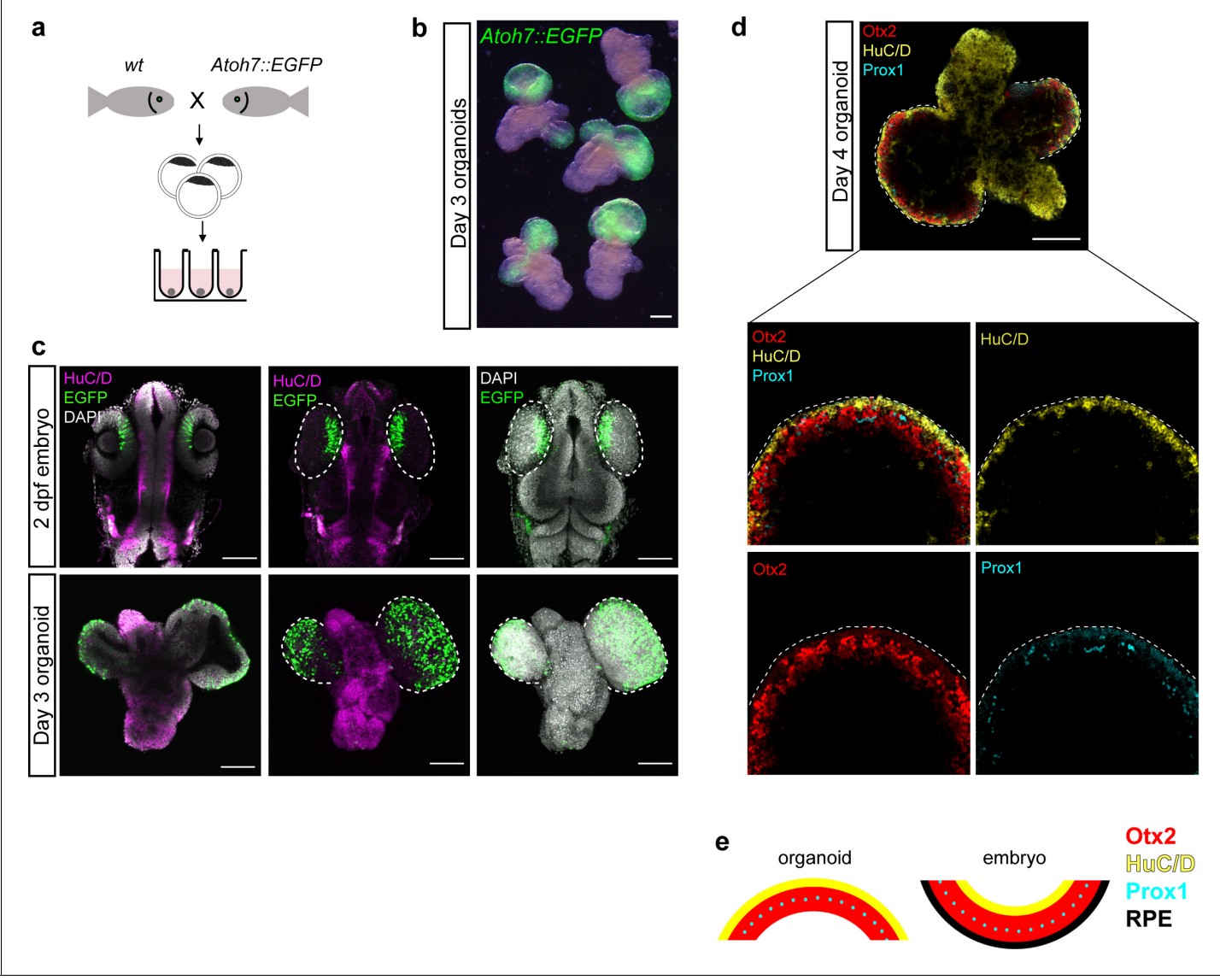

**Figure 6.** Medaka-derived organoids show onset of retinal differentiation. (**a**) Scheme of organoid generation from *Atoh7::EGFP* transgenic fish. (**b**) Fluorescent images of day 3 *Atoh7::EGFP* organoids (generated by aggregation of <1000 cells). (**c**) Optical sections and maximal projections showing EGFP expression in the eye of the developing embryo at 2 dpf and the retinal organoid at day 3 co-stained with antibody against HuC/D and DAPI. (**d**) Optical sections showing expression of HuC/D (amacrine and ganglion cells), Otx2 (bipolar and photoreceptor cells), and Prox1 (horizontal cells) in day 4 organoid. (**e**) Sketch of the arrangement of cellular layers in the organoid and the embryonic retina. dpf, days post-fertilization. Scale bar: 100 μm. The online version of this article includes the following figure supplement(s) for figure 6:

**Figure supplement 1.** Neuroepithelium of the organoids generated by >1000 cells shows the onset on retinal differentiation.

**Figure supplement 2.** Treatment of medaka-derived organoids with the GSK3β inhibitor CHIR-99021 promote RPE differentiation.

with the retinal differentiation potential. By adjusting numbers of aggregating cells and the addition of Matrigel, these aggregates are efficiently directed to the retinal fate, the OV morphogenesis, and the retinal differentiation.

## Discussion

During embryogenesis, individual cells interact with each other and the environment to establish 3D structures of high complexity – tissues, organs, and ultimately the entire embryo. In the absence of immediate access to mammalian embryos, development of 3D in vitro cultures provides a unique

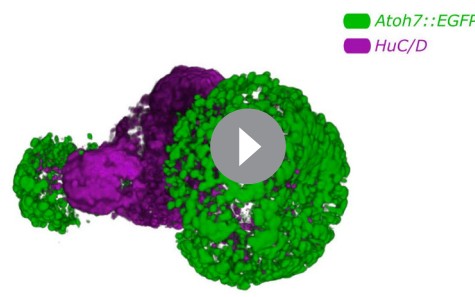

Atoh7::EGFP
HuC/D

**Video 8.** Organoids show complex morphology with the onset of retinal differentiation. 3D rendering of medaka *Atoh7::EGFP* day 3 organoid stained with anti-GFP (green), anti-β-catenin (cyan), and anti HuC/D (magenta) antibodies.
https://elifesciences.org/articles/66998#video8

platform to study development and the roots of pathologies under tightly controlled conditions (*Hofer and Lutolf, 2021*; *Kretzschmar and Clevers, 2016*; *Takebe and Wells, 2019*). In less than a decade, retinal organoids from human and mouse pluripotent stem cells have been shown to recapitulate in vivo retinogenesis, model several retinal diseases, and serve as a potential source for transplantation therapy (*Artero Castro et al., 2019*; *Brooks et al., 2019*; *Fligor et al., 2018*; *Gao et al., 2020*; *Kruczek and Swaroop, 2020*; *Kuwahara et al., 2015*; *Völkner et al., 2016*). While human retinal organoids are the most desired, they also take the longest time to develop and their systematic analysis is limited by technical challenges in genetic manipulations and, last not least, by the lack of direct comparison to the in vivo biological processes. In developmental biology, these challenges are tackled by use of other, more accessible genetic models.

Similarly, the derivation of retinal organoids from non-mammalian models, for example medaka and zebrafish, may provide an easier and more immediate way to address different aspects of eye development. Although fish blastula-derived cells have been used to generate ES cultures (*Ho et al., 2014*; *Hong et al., 1998*; *Hong et al., 1996*; *Peng et al., 2019*; *Robles et al., 2011*; *Yi et al., 2010*), our current understanding of their ability to differentiate and assemble into tissues is very limited. Zebrafish blastula explants have been found to form polarized aggregates capable of specification of all three germ layers (*Fulton et al., 2020*; *Schauer et al., 2020*), resembling mammalian ES cell-derived gastruloids (*Beccari et al., 2018*; *van den Brink et al., 2014*; *Moris et al., 2020*; *Turner et al., 2017*). Besides that, fish primary ES cells have been directed to differentiate to cardiomyocytes forming beating cell sheets with contractile kinetics and electrophysiological features when cultured under defined chemical and mechanical conditions (*Xiao et al., 2016*). However, under no condition reported so far, fish-derived 3D in vitro cultures have formed highly organized neural structures.

Here, we demonstrate that similar to mammalian ES cells, primary embryonic pluripotent cells derived from fish can organize ORGANIZEinto anterior neural structures, particularly the retina, consistent with the proposed default differentiation program followed by pluripotent stem cells (*Eiraku et al., 2011*; *Hemmati-Brivanlou and Melton, 1997*; *Kuwahara et al., 2015*; *Levine and Brivanlou, 2007*; *Nakano et al., 2012*; *Tropepe et al., 2001*; *Turner et al., 2014*). In addition, inter-species blastula cell transplantation experiments between zebrafish and medaka result in ectopic retina formation composed exclusively of donor cells (*Fuhrmann et al., 2020*), further reflecting the intrinsic tendency of primary embryonic pluripotent cells to form retinal structures. Our results show that retinal fate specification in fish-derived organoids is executed spontaneously.

We found that the starting size (cell number) of the initial aggregate crucially impacts on the morphogenesis of the resulting organoid. While aggregates generated from <1000 cells differentiated into anterior neuroectoderm that subsequently underwent morphogenesis, bulging out retinal primordia that closely resembled OVs; aggregates generated from >1000 cells entirely differentiated into a single giant retina. This appeared counterintuitive, since the 'classical' reaction-diffusion model of pattern formation (*Meinhardt, 2012*; *Turing, 1952*) predicts that the number of repeating patterns – OVs – will increase proportionally with tissue size. For fish retinal organoids, the effect is however inverted, offering fish retinal organoids as a model to address alternative pattering scenarios.

Remarkably, re-aggregation of dissociated fish primary pluripotent cells was fast (about 2–3 hr) and highly efficient (100%) when derived from blastula-stage embryos. With OV formation by day 2 and the onset of differentiation by day 3, fish organoids proceed through development almost 10 times faster than their murine counterparts (*Eiraku et al., 2011*; *Kuwahara et al., 2015*; *Nakano et al., 2012*). The formation of fish retinal organoids followed the in vivo developmental timing, reflecting the difference in relative speed of development of the corresponding species.

This rapid development in combination with the ease of highly efficient genome editing (*Gutierrez-Triana et al., 2018*; *Prykhozhij and Berman, 2018*; *Stemmer et al., 2015*), small size, and full transparency granting direct comparison to in vivo processes, pose fish-derived organoids as a rapid and efficient prototyping platform for systematic drug tests in disease models, which then can be translated to more demanding mammalian systems.

Although eye development is highly stereotypic across vertebrates and the players involved are highly conserved, there are certain species-specific features that are also manifested in organoids. One such example is the cellular mechanism of OV evagination. In mammalian eye development, paralleled in mouse and human retinal organoid cultures, the OV is formed by a set of coordinated collective epithelial cell movements driving vesicle evagination from the initially formed neuroepithelium (*Chauhan et al., 2015*; *Eiraku et al., 2011*; *Martinez-Morales et al., 2017*). In teleosts, in contrast, OV formation is driven by the individual migration of *Rx3*-expressing retinal progenitor cells (*Rembold et al., 2006*). Fish-derived retinal organoids exhibit the same mechanism of OV formation, as recapitulated by the migratory trajectories and individual movements of *Rx3*-expressing cells (*Figure 4g,j*). Our analysis indicates that this fundamental property does not depend on the environment (embryo vs. culture) and rather reflects a 'hard-wired' feature of the OV formation.

While the concept of evolutionary conservation allows to deduce fundamental mechanisms by comparison of embryos and the crucially contributing genes, it is not immediately transferable to the comparison of organoids from different species. Selection acts only on development and consequently relies on the entire context of the developing embryo. Morphological and molecular resemblances of organoids to the corresponding organs may indicate a self-organizing capacity that is irrespective of the environment or organismal constraints. The comparison of organoids derived from a wide range of evolutionarily diverse species provides the opportunity to systematically address and distinguish intrinsic and extrinsic constraints contributing to self-organization and organ morphogenesis, and to ultimately tackle the question of the 'conservation' of cellular self-organization, a feature that has apparently not been selected for in evolution.

## Materials and methods

**Key resources table**

| Reagent type (species) or resource | Designation | Source or reference | Identifiers | Additional information |
|---|---|---|---|---|
| Gene (*Oryzia latipes*) | *rx3* | Ensembl | ENSORLG00000027320 | – |
| Strain, strain background (*Oryzia latipes*) | Cab | *Loosli et al., 2000* | – | – |
| Strain, strain background (*Danio rerio*) | AB | ZIRC | ZFIN: ZBD-GENO-960809–7 | – |
| Genetic reagent (*Oryzia latipes*) | *Atoh7::EGFP* | *Del Bene et al., 2007* | – | – |
| Genetic reagent (*Oryzia latipes*) | *Rx3::H2B-GFP* | *Rembold et al., 2006* | – | – |
| Genetic reagent (*Oryzia latipes*) | *Rx3^{saGFP}* | This study | – | Insertion of splice acceptor followed by GFP (saGFP) and ocean pout polyA terminator (OPT) in the first intron of *Rx3* gene |
| Antibody | Anti-Otx2 (polyclonal goat IgG) | R&D Systems | Cat#:AF1979 RRID:AB_2157172 | IHC (1:200) |
| Antibody | Anti-Rx2 (polyclonal rabbit IgG) | In-house; *Reinhardt et al., 2015* | – | IHC (1:200) |
| Antibody | Anti-Lhx2 (polyclonal rabbit IgG) | GeneTex | Cat#:GTX129241 RRID:AB_2783558 | IHC (1:500) |

*Continued on next page*

*Continued*

| Reagent type (species) or resource | Designation | Source or reference | Identifiers | Additional information |
|---|---|---|---|---|
| Antibody | Anti-β-catenin (polyclonal rabbit IgG) | Abcam | Cat#:Ab6302 RRID:AB_305407 | IHC (1:500) |
| Antibody | Anti-acetylated tubulin (monoclonal mouse IgG2b) | Merck | Cat#:T7451 RRID:AB_609894 | IHC (1:200) |
| Antibody | Anti-Sox2 (polyclonal rabbit IgG) | GeneTex | Cat#:GTX124477 RRID:AB_11178063 | IHC (1:200) |
| Antibody | Anti-GFP (polyclonal chicken IgY) | Thermo Fisher Scientific | Cat#:A10262 RRID:AB_2534023 | IHC (1:500) |
| Antibody | Anti-N-cadherin (monoclonal rabbit IgG) | Abcam | Cat#:ab76011 RRID:AB_1310479 | IHC (1:200) |
| Antibody | Anti-Prox1 (polyclonal rabbit IgG) | Merck | Cat#:AB5475 RRID:AB_177485 | IHC (1:500) |
| Antibody | Anti-HuC/HuD (monoclonal mouse IgG2b) | Thermo Fisher Scientific | Cat#:A21271 RRID:AB_221448 | IHC (1:200) |
| Recombinant DNA reagent | pGBT-RP2 (plasmid) | *Clark et al., 2011* | – | Ocean pout polyA terminator (OPT) sequence-carrying plasmid |
| Recombinant DNA reagent | pGGEV_5_linker (plasmid) | Addgene | RRID:Addgene_49285 | – |
| Recombinant DNA reagent | pGGDestSC-ATG (plasmid) | Addgene | RRID:Addgene_49322 | – |
| Recombinant DNA reagent | pGGEV_4_linker (plasmid) | Addgene | RRID:Addgene_49284 | – |
| Recombinant DNA reagent | pGGEV_7'_linker (plasmid) | Addgene | RRID:Addgene_49293 | – |
| Recombinant DNA reagent | Rx3::H2B-GFP (plasmid) | *Rembold et al., 2006* | – | – |
| Recombinant DNA reagent | pGGD(saGFP-OPT-MCS+2) (plasmid) | This study | – | Splice acceptor and GFP (saGFP); ocean pout polyA terminator (OPT) sequence-carrying plasmid |
| Sequence-based reagent | Rx3_F | This study | PCR primer | TCCTTTTTAGACA AATGTGGCTCC |
| Sequence-based reagent | GFP_R | This study | PCR primer | GCTCGACCAGGATGGGCA |
| Sequence-based reagent | pDest_F | This study | PCR primer | ATTACCGCCTTTGAGTGAGC |
| Sequence-based reagent | Rx3_R | This study | PCR primer | GACAGGTATCCG GTAAGCGG |
| Sequence-based reagent | rx3_T1 | This study | sgRNA | AGCAGAGCGCGCAAA GAACC[AGG] |
| Sequence-based reagent | rx3_T2 | This study | sgRNA | AGCGCGCAAAGAACCA GGCA[GGG] |
| Peptide, recombinant protein | Q5 High-Fidelity DNA Polymerase | NEB | Cat#:M0491L | – |
| Peptide, recombinant protein | I-SceI meganuclease | NEB | Cat#:R0694L | – |
| Commercial assay or kit | InnuPREP DOUBLEpure Kit | AnalyticJena | Cat#:845-KS-5050250 | – |
| Chemical compound, drug | CHIR-99021 | Merck | Cat#:SML1046 | – |

*Continued on next page*

*Continued*

| Reagent type (species) or resource | Designation | Source or reference | Identifiers | Additional information |
|---|---|---|---|---|
| Software, algorithm | Geneious R8.1 | Biomatters | – | – |
| Software, algorithm | CCTop | *Stemmer et al., 2015* | RRID:SCR_016890 | – |
| Software, algorithm | Fiji distribution of ImageJ | *Schindelin et al., 2012* | RRID:SCR_002285 | – |
| Software, algorithm | Hyper stack generator (Fiji plugin) | 10.5281/zenodo. 3368134 | – | Generate hyperstack for images acquired with the Acquifer machine |
| Software, algorithm | ElastixWrapper (Fiji plugin) | *Tischer, 2019*; *Klein et al., 2010* | – | – |
| Software, algorithm | MATLAB | MathWorks | RRID:SCR_001622 | – |
| Software, algorithm | MSD analysis | *Tinevez and Herbert, 2020* | – | MATLAB script for MSD analysis |
| Software, algorithm | Directionality analysis | This study https://github.com/ VeneraW/Directionality ANalysisOrganoids (copy archived at swh:1:rev:31a89aead3 e83ac774e13c0161e4 4deafce58f05; *Zilova, 2021*) | – | MATLAB script for organoid directionality analysis |
| Software, algorithm | TrackMate | *Tinevez et al., 2017* | – | – |
| Software, algorithm | Statannot package, Jupyter notebook | https://github.com/ webermarcolivier/ statannot, *Weber, 2020* | – | – |
| Other | Matrigel | Corning | Cat#:356238 | – |
| Other | DRAQ5 | Thermo Fisher Scientific | Cat#:65-0880-92 | 1:1000 |
| Other | CellMask | Thermo Fisher Scientific | Cat#:C10045 | 1:1000 |
| Other | Tissue Freezing Media | Leica | Cat#:14020108926 | – |

## Fish handling and maintenance

Medaka (*O. latipes*) and Zebrafish (*D. rerio*) stocks were maintained according to the local animal welfare standards (Tierschutzgesetz §11, Abs. 1, Nr. 1, husbandry permit AZ35-9185.64/BH, line generation permit number 35–9185.81/G-145/15 Wittbrodt). The following medaka lines were used in this study: Cab strain as a wild type (*Loosli et al., 2000*), *Rx3::H2B-GFP* (*Rembold et al., 2006*), *Atoh7::EGFP* (*Del Bene et al., 2007*), $Rx3^{saGFP}$ (this study). The following zebrafish lines were used in this study: AB zebrafish as a wild type (ZIRC, ZFIN: ZBD-GENO-960809–7).

## Cloning of pGGD(saGFP-OPT-MCS) and generation of $Rx3^{saGFP}$ knock-in line

The ocean pout polyA terminator (OPT) sequence was released from plasmid pGBT-RP2 (*Clark et al., 2011*) via restriction digest with BfaI-FD (Thermo Fisher Scientific), ligated into pGGEV_5_linker (Addgene #49285), and fused into pGGDestSC-ATG (#49322 Addgene) using the Golden GATEway cloning system (*Kirchmaier et al., 2013*) and the following sequences: target site sequence of sgRNA *GFP_T1* (*Stemmer et al., 2015*) inserted in pGGEV_1, a multiple cloning site in pGGEV_2, a strong AD splice acceptor (*Centanin et al., 2011*) fused with a *GFP* variant not targeted by sgRNA *GFP_T1* (*Stemmer et al., 2015*) in pGGEV_3, pGGEV_4_linker (#49284 Addgene), a multiple cloning cassette in pGGEV_6 and pGGEV_7'_linker (#49293 Addgene). Adjustment of open reading frame following the splice acceptor was accomplished via Q5 (NEB, Cat#:M0491L) mutagenesis by inserting a single or two nucleotides (+1 or +2, respectively) to yield gene trap vectors for all three forward frames.

sgRNA *rx3_T1* (AGCAGAGCGCGCAAAGAACC[AGG], PAM in brackets) and *rx3_T2* (AGCGCG-CAAAGAACCAGGCA[GGG], PAM in brackets) were designed using CCTop and cloned and transcribed as described previously (*Stemmer et al., 2015*). The saGFP-OPT-MCS+2 cassette was inserted into the first intron of *Rx3* by non-homologous end joining via microinjection into the cytoplasm of one cell stage medaka zygotes. Injection mix contained 15 ng/µl of each sgRNA *rx3_T1*, *rx3_T2* and *GFP_T1*, 150 ng/µl *Cas9* mRNA and 10 ng/µl pGGD(saGFP-OPT-MCS+2) in nuclease-free water. Embryos were raised and maintained at 28°C in 1× Embryo Rearing Medium (ERM, 17 mM NaCl, 40 mM KCl, 0.27 mM $CaCl_2$, 0.66 mM $MgSO_4$, 17 mM HEPES) and screened for ocular GFP expression 1 dpf on a Nikon SMZ18. Genotyping-PCR was performed with Q5 High-Fidelity DNA Polymerase (NEB) with 98°C initial denaturation for 2 min, followed by 30 cycles of 98°C denaturation for 20 s, 66°C annealing for 30 s, 72°C extension for 25 s. Primers used to amplify 5′ integration: Rx3_F 5′-TCCTTTTTAGACAAATGTGGCTCC, GFP_R 5′-GCTCGACCAGGATGGGCA; 3′ integration: pDest_F 5′-ATTACCGCCTTTGAGTGAGC, Rx3_R 5′-GACAGGTATCCGGTAAGCGG. Following gel electrophoresis, amplicons were gel-purified (InnuPrep, AnalyticJena), sequenced (Eurofins Genomics), and analyzed using Geneious R8.1 (Biomatters).

## Injection in zebrafish embryos
Rx3::H2B-GFP DNA (10 ng/µl) was co-injected with Meganuclease (I-SceI) (NEB, Cat#:R0694L) and *mCherry* mRNA (10 ng/µl) into the cytoplasm of one cell stage zebrafish embryos as previously described (*Thermes et al., 2002*). Only morphologically intact and brightest mCherry expressing embryos were used for aggregate formation.

## Generation of organoids
For medaka, blastula-stage (6 hpf) embryos (*Iwamatsu, 2004*) were collected, dechorionated using hatching enzyme, and washed in ERM (17 mM NaCl, 40 mM KCl, 0.27 mM $CaCl_2$, 0.66 mM $MgSO_4$, 17 mM HEPES). Cell mass was separated from yolk, washed three times with sterile PBS (Thermo Fisher Scientific, Cat#:10010023), and dissociated by gentle pipetting with 200 µl pipet tip. Cell suspension was pelleted (180 × g for 3 min) and re-suspended in differentiation media: GMEM (Glasgow's Minimal Essential Medium, Gibco Cat#:11710035), 5% KSR (Gibco Cat#:10828028), 0.1 mM non-essential amino acids, sodium pyruvate, 0.1 mM β-mercaptoethanol, 50 U/ml penicillin-streptomycin to desired cell density, that is, for aggregation of >1000 cells, 10–20 cells/µl and for aggregation of <1000 cells, 5–8 cells/µl. Cell suspension (100 µl per well – per single aggregate) was transferred to low binding 96-well plate (Nunclon Sphera U-Shaped Bottom Microplate, Thermo Fisher Scientific) and centrifuged (for 3 min at 180 × g) to speed up cell aggregation. The aggregates were incubated over night at 26°C in the incubator without $CO_2$ control. The following day (day 1) aggregates were washed with differentiation media, transferred to fresh wells, and Matrigel (Corning, Cat#:356238) was added to the media to a final concentration of 2%. From day 1 onward, aggregates were incubated at 26°C and 5% $CO_2$. Alternatively, when $CO_2$ control was not possible, media was supplemented with 20 mM HEPES, pH=7.4. From day 2 onward, organoids were kept in DMEM/F12 supplemented with 5% FBS (Sigma Aldrich, Cat#:12103C), 5% FEE (fish embryonic extract) (https://zfin.org/zf_info/zfbook/chapt6.html), 20 mM HEPES pH=7.4, N2 supplement (Gibco, Cat#:17502048) and 50 U/ml penicillin-streptomycin.

For zebrafish cell aggregation, blastula-stage embryos (at high stage) were processed according to the same protocol as for medaka but re-suspended in Leibowitz's L-15 media (Gibco, Cat#:11415064) supplemented with 5% KSR and 50 U/ml penicillin-streptomycin. Matrigel (Corning, Cat#:356238) was added at 2 hpa.

## Treatment of medaka organoids with GSK3 inhibitor CHIR-99021
After 6 hr of incubation with Matrigel at 26°C, day 1 aggregates were washed and supplemented with differentiation media containing 5 µM CHIR-99021 (Merck, Cat#:SML1046) in DMSO or DMSO only and incubated till the point of analysis (day 2 or day 4).

## Fluorescent labeling

To stain the plasma membranes, organoids were incubated in CellMask Orange Plasma Membrane Stain (Thermo Fisher Scientific, Cat#:C10045; 1:1000) for 30 min at 26°C, washed and fixed in 4% PFA.

Immunohistochemistry was performed as previously described (*Inoue and Wittbrodt, 2011*) with slight modifications. Embryos and organoids were fixed in 4% PFA overnight at 4°C, and washed in PTW (PBS with 0.05% Tween20). For sectioning, fixed samples were cryopreserved in 30% (w/w) sucrose over night at 4°C. Samples were equilibrated in the 1:1 mixture of sucrose and Tissue Freezing Media (Leica; Cat#:14020108926) overnight at 4°C, frozen in Tissue Freezing Media and sectioned to 10–12 µm. Sections were re-fixed for 10 min in 4% PFA, washed in PTW, blocked for 2 hr in 10% BSA, incubated with primary antibody (anti-Otx2, R&D Systems, Cat#:AF1979) overnight and washed 5 times 10 min in PTW. After 2 hr incubation with secondary antibody, samples were washed in PTW, co-stained with DAPI nuclear stain (1:500), mounted in 60% glycerol, and imaged with Leica Sp8 confocal microscope.

For whole-mount staining, fixed samples were heated in 50 mM Tris-HCl at 70°C for 15 min, permeabilized 15 min in acetone at –20°C, and blocked in 10% BSA in PTW for 1 hr. Samples were incubated with primary antibody (1:200) overnight (for Otx2 antibody incubation was prolonged to 3 days) at 4°C. The following antibodies were used: rabbit anti-Rx2 (*Reinhardt et al., 2015*), rabbit anti-Lhx2 (GeneTex, Cat#:GTX129241), rabbit anti-β-catenin (Abcam, Cat#:Ab6302), mouse anti-acetylated tubulin (Merck, Cat#:T7451), rabbit anti-Sox2 (GeneTex, Cat#:GTX124477), chicken anti-GFP (Thermo Fisher Scientific, Cat#:A10262), goat anti-Otx2 (R&D Systems, Cat#:AF1979), rabbit anti-N-cadherin (Abcam, Cat#:ab76011), rabbit anti-Prox1 (Merck, Cat#:AB5475), and mouse anti-HuC/D (Thermo Fisher Scientific, Cat#:A21271). Samples were washed six times 10 min in PTW, incubated with secondary antibody (1:750) (Invitrogen) with DAPI (1:500) or DRAQ5 (Thermo Fisher Scientific, Cat#:65-0880-92; 1:1000) nuclear stain overnight at 4°C and washed five times 10 min in PTW. All samples were mounted in 1% low melting agarose in PTW and imaged with Leica Sp8 confocal, Acquifer, SPIM microscopes.

## Imaging

Fixed organoid and embryonic samples were imaged with Leica Sp8 confocal microscope. Gross morphology of embryos and organoids was assessed by Nikon SMZ18 and Leica DMi8 microscope.

Time-lapse imaging of aggregation and *Rx3* expression profiles in organoids (*Videos 1–4*) was performed on the ACQUIFER Imaging Machine (ACQUIFER Imaging GmbH, Heidelberg, Germany) (*Pandey et al., 2019*). Aggregates, single cells, or dechorionated embryos were loaded into 96-well plates and placed in the plate holder of the ACQUIFER machine at 26°C for medaka and 28°C for zebrafish. For each well plate, a set of 10 z-slices (75 µm step size) was acquired in the bright field (50% LED intensity, 50 ms exposure time) and 470 nm fluorescence (50% LED excitation source, FITC channel, 200 ms exposure time) channels with a 4× NA 0.13 objective (Nikon, Düsseldorf, Germany). Imaging was performed over 20 hr of organoid development with 30 min intervals.

3D imaging of fixed *Atoh7::EGFP*-derived organoids (*Video 8*) and N-cadherin and DRAQ5-stained day 2 organoids (*Figure 1—figure supplement 2*) was performed on multiview selective-plane illumination MuVi SPIM Multiview light-sheet microscope (Luxendo Light-sheet, Bruker Corporation) (*Krzic et al., 2012*). The organoids were mounted in 1% low melting agarose (StarPure Low Melt Agarose, Cat#N3103-0100, StarLab GmbH) inside an FEP tube (Karl Schupp AG) fixed on glass capillaries. Four volumes (two cameras and two rotation angles) were acquired with the 25× detection setup (*Caroti et al., 2018*), 1 µm z step size for three channels: GFP (Atoh7) 488 nm excitation laser at 40% intensity, 525/50 nm emission filter, 600 ms exposure time; RFP (β-catenin) 561 nm excitation laser at 40% intensity, 607/70 nm emission filter, 600 ms exposure time; far Red (HuC/D) 642 nm excitation laser at 50% intensity, 579/40 nm emission filter, 1000 ms exposure time. The volumes were fused with Luxendo Image fusion software. Fused volumes were visualized by 3D rendering with Amira 6.2 Main software.

Live imaging of *Rx3::H2B-GFP*-derived organoids (*Video 5*) was performed on the 16× detection MuVi SPIM Multiview light-sheet microscope (Luxendo Light-sheet, Bruker Corporation). To assure normal development of organoids, the FEP tube was partially filled with 2% low melting agarose. After solidification of agarose, the tube was further filled with differentiation media and an organoid

was scooped inside the tube. The tube was positioned vertically such that the organoid fell onto the solid support of agarose. Two volumes with 1.6 µm z step size for two channels, that is, GFP 488 nm excitation laser at 10% intensity, 525/50 nm emission filter, 50 ms exposure time and quasi bright field 561 nm excitation laser at 5% intensity, 568LP nm emission filter, 100 ms exposure time, were acquired every 15 min.

Live imaging of *Rx3::H2B-GFP*-derived organoids (*Videos 6* and *7*) was performed with Sp8 confocal microscope (Leica). Organoids were imaged from day 1 to day 2 at room temperature directly in low binding 96-well plate (Nunclon Sphera U-Shaped Bottom Microplate, Thermo Fisher Scientific). The volume of the organoids was acquired with 2.24 µm z step size using 488 nm excitation laser at 10% intensity in 30 min intervals.

## Quantitative analysis

To analyze the dynamics of retinal cell fate acquisition, for each time point, a maximum projection for fluorescence and a single focused bright-field image were calculated using Fiji distribution of ImageJ (*Schindelin et al., 2012*) and hyper stack plugin for ACQUIFER machine (https://doi.org/10.5281/zenodo.3368134). The onset of GFP expression was determined by detection of maximum in first derivative of GFP expression. For better representation, the sum of fluorescence intensity for each time point was normalized from 0 to 1.

Analysis of gene expression in <1000 cell and >1000 cell organoids was performed on images acquired with Leica DMi8 microscope. Automatic thresholding (minimum) on bright channel was used to determine area of an organoid and on fluorescence image to identify areas corresponding to Rx3 or Rx2 expression.

To analyze cell migration behavior in organoids of day 2, time-lapse volumes acquired with SPIM were registered with ElastixWrapper for Fiji (*Tischer, 2019*; *Klein et al., 2010*) using Euler transformation, 1000 iterations, full data points. The cells within organoids were tracked with TrackMate (*Tinevez et al., 2017*), using simple linear tracker. The analysis of tracks' directionality was performed with custom-written MATLAB script (https://github.com/VeneraW/DirectionalityAnalysisOrganoids; *Zilova, 2021*). The MSD analysis including velocity autocorrelation was performed with MATLAB tool for analysis of particle trajectories (*Tinevez and Herbert, 2020*). For demonstration purposes, we chose a few representative tracks with high total displacement (total distance traveled) and duration (time required for travel).

Statistical analysis and plots were prepared with Jupyter notebook (*Perez and Granger, 2007*), using statannot package (https://github.com/webermarcolivier/statannot) to compute statistical tests (Wilcoxon-Mann-Whitney) and add statistical annotations. For all figures, ns $0.05<p<1$, *$0.01<p<0.05$, **$0.001<p<0.01$, ***$0.0001<p<0.001$, ****$p<0.0001$. Figures were assembled with Adobe Illustrator CS6.

## Acknowledgements

The authors thank L Centanin, E Tsingos, N Sokolova, and Z Kozmik for valuable discussions and comments on the manuscript. We are grateful to I Thomas, A Sarvari, M Pandya, and N Priya for the help at the initial stage of the project. We thank Darius Balciunas for providing the GBT-RP2 plasmid containing the OPT sequence. VW was supported by the German Research Foundation research fellowship WE 6221/2–1. This work was supported by grants of the Excellence Cluster '3D Matter Made to Order' (3DMM2O) funded through the German Excellence Strategy via Deutsche Forschungsgemeinschaft (DFG), by the Carl Zeiss Foundation and by the ERC Synergy Grant IndiGene (Number 810172) to JW.

## Additional information

### Funding

| Funder | Grant reference number | Author |
| --- | --- | --- |
| H2020 European Research Council | 810172 | Joachim Wittbrodt |

| Deutsche Forschungsge-meinschaft | 3DMM2O | Joachim Wittbrodt |
| Deutsche Forschungsge-meinschaft | WE 6221/2-1 | Venera Weinhardt |
| Carl Zeiss Foundation | HeiKa Graduate School | Christina Schlagheck |

The funders had no role in study design, data collection and interpretation, or the decision to submit the work for publication.

## Author contributions
Lucie Zilova, Conceptualization, Data curation, Formal analysis, Supervision, Validation, Investigation, Visualization, Methodology, Writing - original draft; Venera Weinhardt, Conceptualization, Data curation, Software, Formal analysis, Supervision, Validation, Investigation, Visualization, Methodology, Writing - original draft; Tinatini Tavhelidse, Thomas Thumberger, Resources, Methodology, Writing - review and editing; Christina Schlagheck, Formal analysis, Investigation; Joachim Wittbrodt, Conceptualization, Resources, Supervision, Funding acquisition, Writing - original draft, Project administration

## Author ORCIDs
Lucie Zilova (iD) https://orcid.org/0000-0001-6404-9119
Venera Weinhardt (iD) https://orcid.org/0000-0002-9774-3833
Tinatini Tavhelidse (iD) https://orcid.org/0000-0002-6103-9019
Christina Schlagheck (iD) https://orcid.org/0000-0002-1311-5945
Thomas Thumberger (iD) http://orcid.org/0000-0001-8485-457X
Joachim Wittbrodt (iD) https://orcid.org/0000-0001-8550-7377

## Decision letter and Author response
Decision letter https://doi.org/10.7554/eLife.66998.sa1
Author response https://doi.org/10.7554/eLife.66998.sa2

# Additional files
## Supplementary files
• Transparent reporting form

## Data availability
Raw datasets from time-lapse experiments (Figure 5, Video 5) are deposited in publicly available repository HeiData (https://heidata.uni-heidelberg.de/).

The following dataset was generated:

| Author(s) | Year | Dataset title | Dataset URL | Database and Identifier |
|---|---|---|---|---|
| Weinhardt V, Zilova L, Tavhelidse T, Schlagheck C, Thumberger T, Wittbrodt J | 2021 | Fish primary embryonic pluripotent cells assemble into retinal tissue mirroring in vivo early eye development - in vivo imaging of OV formation | https://doi.org/10.11588/data/AOSUS8 | Heidelberg University (HeiData), 10.11588/data/AOSUS8 |

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
