## [Decision Letter]

**Acceptance summary:**

In this manuscript, Zilova et al. show that primary embryonic cells derived from blastula-stage Medaka and Zebrafish embryos can self-organize into retinal organoids. This is a novel and original piece of work that reveals the capacity of fish primary embryonic pluripotent cells to behave like mammalian embryonic stem cells and organize optic cup organoids.

**Decision letter after peer review:**

Thank you for submitting your article "Fish primary embryonic stem cells self-assemble into retinal tissue mirroring in vivo early eye development" for consideration by *eLife*. Your article has been reviewed by 3 peer reviewers, including Alfonso Martínez Arias as the Reviewing Editor and Reviewer #1. and the evaluation has been overseen by Didier Stainier as the Senior Editor.

Essential revisions:

1) The introduction is very sparse and should be expanded. At the moment it reads as it had been cut off. A more thorough introduction of the issues will help highlighting the novelty of their observations. In it they should refer to PMID 17540356 and, in particular Figure 2 and references therein about old newt experiments hinting at what they observe here. They should also avoid referring to their cells as 'stem cell', they are (as they say elsewhere) primary embryonic pluripotent cells.

2) It would be helpful to have a timeline of retinal development at the beginning of the paper (figure 1). In this way, it will be easier for the reader to follow the figures in the paper that go from the neuroepithelium to the optic-vesicle like structures in the organoids.

3) In Figure 1d, they should show markers of epithelial cells e.g Cadherins or ZO-1.

4) In Figure 2, It is important to show Rx3 expression before Matrigel embedding. It seems to be polarized but if this is the case, it is important to show the frequency of this event. It would also be good to have a video of the development of the organoids in Matrigel which, at the moment, is missing (or we have not found it). It is important to show the transition from 'polarised' Rx3 expression in the organoids go to the Rx3 expression in the neuroepithelium?

5) In Figure 4, the authors should specify that this is a Medaka experiment. Figure 4a is very small. It would be important to have a large view of the organoids and the existent variability in these.

6) In the smaller organoids (figure 4a, e, g), is there epithelium formation similar to the regular organoids? Because it does not seem to. So, is epithelium necessary to progress to retinal differentiation?

7) In Figure 4, the authors show that the optic vesicle organoids are organized as in vivo with cells expressing RPE markers. These cells are no longer present in Figure 5. What happens to them? There is no mention of this problem in the text. This should be addressed or a least discussed. The RPE absence may be a reason why the retina differentiates with an inverted organization.

8) The authors variably interpret their observations as the result of self-assembly or self-organization. At the moment, the data does not allow distinguishing whether the observed phenomena result from cells following largely cell-autonomous differentiation paths and come together through cell sorting, or whether dissociation and aggregation generates a condition that leads to (spatially restricted) retinal differentiation in cells that would not normally adopt this fate. I would say that the first scenario is consistent with self-assembly, while the second one is more self-organized in the sense that the new cell-cell interactions resulting from the aggregation result in emergent cellular behaviours. A first step to distinguish between these possibilities would be to quantitatively demonstrate that aggregation biases cell differentiation towards neural and retinal fates at the expense of other cell types, compared to the intact embryo. The examples shown in Figure 2 and 3d seem to indicate an overrepresentation of neural cells, but it would be good to see a quantitative comparison to the embryo.

9) The authors claim that their system is highly reproducible. Unfortunately, they do not give an indication of the success rate of aggregate formation in figure 1. Figure 4 shows the most complex patterns, but I realize that there is quite a bit of variability in between the aggregates – they are just as likely to have one or two Rx2-expressing areas (panel b). I also could not find information how many aggregates show the patterns in panels e and f, and from how many aggregates the data in panels g – i has been collected.

*Reviewer #1 (Recommendations for the authors):*

In this manuscript, Zilova et al. show that primary embryonic cells derived from blastula-stage Medaka and Zebrafish embryos can self-organize into retinal organoids. When aggregates of 1000-2000 primary embryonic cell are embedded in Matrigel addition, they form a neuroepithelium under the control of Rx3 which develops into a retinal organoid. The process mirrors some aspects of embryo development. Moreover, another interesting finding is that Rx3 expression is initiated in the absence of Matrigel at day 0, which indicates that the retinal fate occurs by default and is not dependent on extracellular matrix components. The authors compare the ability of cells from Mesaka and zebra fish and show that both are competent to form organoids, though each does it with the time scale of the embryo of origin. The authors show that by reducing the number of Medaka cells to aggregate (500-800 cells), Rx2 and Rx3 are expressed only in restricted regions of the small aggregates, presumably where they organize into discrete circular Rx2 and Rx3 positive neuroepithelial units that develop into structure resembling retinal epitjhelia with some diversity of retinal cell types including amacrine, ganglion, photoreceptor, bipolar and horizontal cells.

This is a novel and original piece of work that reveals the capacity of fish primary embryonic pluripotent cells to behave like mammalian embryonic stem cells and organize optic cup organoids. It is worth considering it for publication but first the authors should address a number of issues that could improve the presentation of their findings and also expand their observations beyond the description of the phenomenon.

Details are indicated below but there are two pieces of additional information that would unwrap the potential of their observations. The first one is the, these days almost obligatory, single cell analysis of their retinal organoids compared to the embryo, particularly of the retinal epithelium. Of course, were they able to deepen the analysis in Figure 5 this would not be necessary, but a more complete account of the cell types present in the organoids is necessary as is whether (or how much brain tissue there is). In addition, the observations of the different timings of the ex vivo development of Medaka and zebrafish begs the experiment of doing the experiment of generating chimeric organoids and see how the combinations of cells from different species affects their development. A more detailed characterization of the development of the organoids in comparison with the embryo and, importantly, a detailed reporting of the frequency of the events, will be appreciated.

The present manuscript also requires clarification of several points

The introduction is very sparse and should be expanded. At the moment it reads as it had been cut off. A more thorough introduction of the issues will help highlighting the novelty of their observations. In it they should refer to PMID 17540356 and, in particular Figure 2 and references therein about old newt experiments hinting at what they observe here. They should also avoid referring to their cells as 'stem cell', they are (as they say elsewhere) primary embryonic pluripotent cells.

It would be helpful to have a timeline of retinal development at the beginning of the paper (figure 1). In this way, it will be easier for the reader to follow the figures in the paper that go from the neuroepithelium to the optic-vesicle like structures in the organoids.

Other issues they should address:

At the beginning, they should not refer to the structures as 'organoids' at this stage as they are only epithelial cysts at this point.

In Figure 1, they should discuss a comparison of figures 1c, d with figure 4a, c. Are these hollow in the middle or do they have cells? Is the middle GFP expression autofluorescence?

In Figure 1d, they should show markers of epithelial cells e.g Cadherins or ZO-1.

In Figure 2, It is important to show Rx3 expression before Matrigel embedding. It seems to be polarized but if this is the case, it is important to show the frequency of this event. It would also be good to have a video of the development of the organoids in Matrigel which, at the moment, is missing (or we have not found it). It is important to show the transition from 'polarised' Rx3 expression in the organoids go to the Rx3 expression in the neuroepithelium?

Figure 2g – It would be easier to read and understand by depicting Rx3-/- for the mutants rather than Rx3::saGFP+/+.

On the zebrafish experiments, Figure 3, need to state in-text the exact stage of the zebrafish from which the organoids where then generated. In methodology is stated: ' For zebrafish cell aggregation, blastula-stage embryos were processed according to the same protocol and re-suspended in Leibowitz's L-15 media supplemented with 5% KSR and penicillin-streptomycin. Matrigel (Corning, 356238) was added 2 hpa'. But the authors should point the exact number of cells here. How many where they added? Have the authors tested aggregating different number of cells as with Medaka and examined Rx3 expression and morphogenesis? Important to have more comparisons with the medaka.

It would also be helpful to depict the protocol of the organoids generation from Zebrafish in a schematic as done with the Medaka. And indicate until which time point and developmental stage can they progress. Do we see similar retinal differentiation as with the medaka (figure 5)?

Compare Figure 3b with 3d: Looking at the organoids, one can see major differences here. The organoid shown at figure 3b is circular with an outer neuroepithelium (assuming also that Rx3 expression is circular as with medaka) whereas organoids in figure 3d are variable with Rx3 expression being polarised. Also, again is this here neuroepithelium? It would be important to stain for Cadherin.

In Figure 4, the authors should specify that this is a Medaka experiment. Figure 4a is very small. It would be important to have a large view of the organoids and the existent variability in these.

In Figure 4c it is stated that the 100% of the regions of regular organoids express Rx2. But in figure 1d we can see that the center might be hollow. So, again, is there autofluorescence in figure 4a (also see figure 2g) or not? Need to clarify.

Figure 4e: how many organoids show this pattern of gene expression? Is there variability when we have 2 poles of Rx3 expression? The issue of frequencies is very important in this field and the authors should report it.

In the smaller organoids (figure 4a, e, g), is there epithelium formation similar to the regular organoids? Because it does not seem to. So, is epithelium necessary to progress to retinal differentiation?

In Figure 5, compare figure 5b with 5c: One can see organoids with one pole of Atoh7. So, in figure 7c how often is this differentiation phenomenon?

Figure 5d: need to show an embryo to compare with the organoids' results.

*Reviewer #2 (Recommendations for the authors):*

In Figure 4, the authors show that the optic vesicle organoids are organized as in vivo with cells expressing RPE markers. These cells are no longer present in Figure 5. What happens to them? There is no mention of this problem in the text. This should be addressed or a least discussed. The RPE absence may be a reason why the retina differentiates with an inverted organization.

The discussion is generally informative but somehow fails to provide real advantages of using teleost organoids vs the fish per se or vs for example human organoids. Indeed, obtaining a fish organoid is faster that a human one, but more expensive and time consuming than using fish embryos. The author should perhaps provide more information in the introduction to explain what has push them to undertake this effort, besides the technical challenge. What can we learn from fish vs mammalian organoids?

*Reviewer #3 (Recommendations for the authors):*

1) It was not immediately obvious to me what motivated the authors to study retinal differentiation in the aggregates. (line 95 ff). This could be discussed more clearly.

2) It is unclear to me what the quantity "level of organization" in figure 2h means. Is there a more specific way to express this?

3) I was a bit confused with the positioning of the zebrafish experiments in the manuscript. It took me a while to realize that the experiments in Figure 4 were performed in the medaka system again. Perhaps the zebrafish data could be discussed elsewhere in the manuscript, or the authors could indicate more clearly in the text which system they used for the respective experiments.

4) The discussion dwells extensively on potential uses of the fish aggregate system as an alternative to organoids from mammalian cells. I am not convinced that this is a strong point, as potential advantages due to faster development are offset by evolutionary differences. Rather, I think that these systems are of interest in their own to investigate developmental mechanisms of cell differentiation and morphogenesis.

5) The reference to Fuhrmann et al., 2020 in line 389 seems to be wrong.

6) I find the last section of the discussion hard to grasp. If the authors are referring to previously described concepts, it might be helpful to add some references here.

---

## [Author Response]

Essential revisions:1) The introduction is very sparse and should be expanded. At the moment it reads as it had been cut off. A more thorough introduction of the issues will help highlighting the novelty of their observations. In it they should refer to PMID 17540356 and, in particular Figure 2 and references therein about old newt experiments hinting at what they observe here. They should also avoid referring to their cells as 'stem cell', they are (as they say elsewhere) primary embryonic pluripotent cells.

Indeed, the introduction was very concise. We happily responded to the suggestion of the referees and have extended the introduction and have taken care to incorporate and quote all the suggested work. Throughout the revised version of the manuscript we have taken care to consistently refer to the “starting material” as “primary embryonic pluripotent cells”.

2) It would be helpful to have a timeline of retinal development at the beginning of the paper (figure 1). In this way, it will be easier for the reader to follow the figures in the paper that go from the neuroepithelium to the optic-vesicle like structures in the organoids.

We followed the referee’s suggestion and have included a scheme of retina development in Figure 1 (panel a) for reference and introduce retina development in the extended introduction of the revised version of the manuscript.

3) In Figure 1d, they should show markers of epithelial cells e.g Cadherins or ZO-1.

We repeated the respective analyses and have now included N-cadherin staining on day 2 organoids presented in the new Figure 1 panel d, as well as in the linked supplements (Figure 1—figure supplement 2) and Figure 2 panel e.

4) In Figure 2, It is important to show Rx3 expression before Matrigel embedding. It seems to be polarized but if this is the case, it is important to show the frequency of this event. It would also be good to have a video of the development of the organoids in Matrigel which, at the moment, is missing (or we have not found it). It is important to show the transition from 'polarised' Rx3 expression in the organoids go to the Rx3 expression in the neuroepithelium?

To address the question concerning distribution of *Rx3*-expressing cells in relation to the addition of Matrigel, we have performed extended time-lapse imaging on all organoids (n=54) of a single batch, see video 2 (new panel c, Figure 2). Our analysis shows that there is no polarisation of *Rx3* expression prior to the addition of Matrigel. *Rx3* expression emerges as salt and pepper pattern and only subsequently continuous *Rx3* expressing domains emerge as highlighted in the revised Figure 2. We have also included panels addressing *Rx3* expression before and after Matrigel addition in panel e, Figure 2 showing the distribution of *Rx3*-expressing cells in the context of forming neuroepithelium.

Additionally, we included the data showing the impact of Matrigel on the formation of the neuroepithelium (Figure 2—figure supplement 2). Furthermore, Video 7 addresses the behaviour of *Rx3*-expressing cells in the presence or absence of Matrigel in a time frame of more than 28 hours. Videos 5 and 6 show behaviour of *Rx3*-expressing cells after Matrigel addition up to optic vesicle formation.

The referees had referred to an apparent polarisation of one organoid presented in figure 2 panel c of the initially submitted manuscript. This panel was misleading and as stated above we have in fact not observed any polarized *Rx3* expression. On the contrary, *Rx3* expression emerged in individual, isolated cells in a salt and pepper fashion. We have taken care to avoid the impression of polarized *Rx3* expression (due to maximum projection and high gain) in the revised version, both in the figure as well as in the video.

With the addition of Matrigel and following epithelialisation, a continuous *Rx3* expression domain is forming.

We have refined our statements following the feedback of the referees in the revised version of the manuscript.

5) In Figure 4, the authors should specify that this is a Medaka experiment. Figure 4a is very small. It would be important to have a large view of the organoids and the existent variability in these.

We have revised Figure 4 to include large overview images for panel a and Figure 4—figure supplement 1.

Both figures are excellent examples of the morphological variability of organoids generated by the aggregation of less than 1,000 cells.

For a better representation of the variable number of optic vesicles forming, apparent as evaginated *Rx3*-positive retinal regions, we have revised Figure 4 and present an improved panel c presenting the absolute number of organoids with 1, 2, 3 or 4 optic vesicles/retinal regions.

This allows to instantly appreciate the impact of the starting cell number on the number of forming optic vesicles/retinal regions.

Details on the morphological variability of 72 organoids derived from less than 1,000 cells are now presented in Figure 4—figure supplement 1.

6) In the smaller organoids (figure 4a, e, g), is there epithelium formation similar to the regular organoids? Because it does not seem to. So, is epithelium necessary to progress to retinal differentiation?

The formation of an epithelium does not depend on the number of starting cells. We have addressed this question, in Figure 6—figure supplement 1. Here we show that the onset of retinal differentiation in organoids does not depend on the starting size of the initial aggregate. There is no difference in the onset of retinal differentiation (analysed by the expression of *Atoh7::EGFP*) and layering of retinal cells. Although organoids generated by aggregation of > 1,000 and < 1,000 cells show different morphologies and distribution of retinal domains/optic vesicles, the forming retinal neuroepithelium follows the process or retinal differentiation irrespective of these morphological differences. Epithelium formation (in both, > 1,000 and < 1,000 organoids) is a prerequisite for the survival of organoids as is now presented in Figure 2—figure supplement 2.

7) In Figure 4, the authors show that the optic vesicle organoids are organized as in vivo with cells expressing RPE markers. These cells are no longer present in Figure 5. What happens to them? There is no mention of this problem in the text. This should be addressed or a least discussed. The RPE absence may be a reason why the retina differentiates with an inverted organization.

The referees relate to an important point: As in human and mouse retinal organoids, also in fish retinal organoids the differentiation of the neuroretina is favoured over the formation of RPE. In established systems, this can be induced by supplementing the culture media with Wnt/β-catenin pathway-inducing agents (Eiraku et al., 2011; Nakano et al., 2012; Kuwahara et al., 2015). We demonstrate that this approach also induces RPE in fish-derived organoid. In the revised version of the manuscript we have included our results on RPE differentiation by treatment with the canonical Wnt/β-catenin pathway agonist CHIR99021, see Figure 6—figure supplement 2 and discussed in the manuscript on page 21.

In the absence of a shielding RPE, ECM components provided by the addition of Matrigel polarizes the forming epithelium in an “inverted” fashion.

8) The authors variably interpret their observations as the result of self-assembly or self-organization. At the moment, the data does not allow distinguishing whether the observed phenomena result from cells following largely cell-autonomous differentiation paths and come together through cell sorting, or whether dissociation and aggregation generates a condition that leads to (spatially restricted) retinal differentiation in cells that would not normally adopt this fate. I would say that the first scenario is consistent with self-assembly, while the second one is more self-organized in the sense that the new cell-cell interactions resulting from the aggregation result in emergent cellular behaviours. A first step to distinguish between these possibilities would be to quantitatively demonstrate that aggregation biases cell differentiation towards neural and retinal fates at the expense of other cell types, compared to the intact embryo. The examples shown in Figure 2 and 3d seem to indicate an overrepresentation of neural cells, but it would be good to see a quantitative comparison to the embryo.

Here the referees open a debate that is controversial in the field and we are happy to contribute. We followed the referees’ suggestions on how to address the difference between self-assembly and self-organization and are providing two additional experiments tackling this point.

To address whether the cells would autonomously follow the retinal differentiation path we dissociated blastulae and cultured individual dissociated blastula-stage cells (*Rx3::H2B-GFP* reporter line) in suspension or as a single cells.

Our results added as Figure 2—figure supplement 1 show that, consistent with a default neuronal state, individual blastula-derived cells stochastically acquire retinal fate autonomously, in the absence of cell-cell contacts established by cell aggregation. The clones derived from individual cells eventually show a distribution of retinal (Rx3+) and non-retinal cells.

In the absence of a fully conclusive answer for either self-organization or self-assembly (or likely a combination of both) we have taken care not to confuse the reader by an imprecision in terminology in the crucial point. In light of the historical debate about the differences between self-assembly and self-organisation “Self-assembly, Self-Organization: A philosophical perspective on a major challenge of nanotechnology” by Bensaude-Vincent https://halshs.archives-ouvertes.fr/halshs-00350831/document, we have revised our manuscript accordingly.

9) The authors claim that their system is highly reproducible. Unfortunately, they do not give an indication of the success rate of aggregate formation in figure 1. Figure 4 shows the most complex patterns, but I realize that there is quite a bit of variability in between the aggregates – they are just as likely to have one or two Rx2-expressing areas (panel b). I also could not find information how many aggregates show the patterns in panels e and f, and from how many aggregates the data in panels g – i has been collected.

The reviewer touches a crucial point and actually one to the strengths of our system. Overall the derivation of retinal organoids from primary embryonic pluripotent cells is highly reproducible (in 100% of all experiments performed) with routinely almost 100 % efficiency per experiment (video 1 and video 2). We clearly state and present those key aspects in the revised version of the manuscript.

With our protocol on derivation of organoids, all combined primary embryonic pluripotent cells proceed through aggregation (84 out of 84 aggregates in video 1 and 2). In an independent experiment we show the highly synchronous onset of *Rx3* expression (54 out of 54 aggregates in video 2). There is a variability in size (panel d Figure 4) and number of optic vesicles (panel c Figure 4 and Figure 4—figure supplement 1), all of which will eventually initiate retinal differentiation (panel b Figure 6—figure supplement 1). To allow the reader to get an immediate impression of the efficiency of the procedure, we have now added the number of organoids used in every experiment in figure captions.